# Towards characterizing the value of edge embeddings in Graph Neural Networks

**Dhruv Rohatgi** [1]  **Tanya Marwah** [2]  **Zachary Chase Lipton** [2]  **Jianfeng Lu** [3]  **Ankur Moitra** [1]  **Andrej Risteski** [2]

## Abstract

Graph neural networks (GNNs) are the dominant approach to solving machine learning problems defined over graphs. Despite much theoretical and empirical work in recent years, our understanding of finer-grained aspects of architectural design for GNNs remains impoverished. In this paper, we consider the benefits of architectures that maintain and update edge embeddings. On the theoretical front, under a suitable computational abstraction for a layer in the model, as well as memory constraints on the embeddings, we show that there are natural tasks on graphical models for which architectures leveraging edge embeddings can be much shallower. Our techniques are inspired by results on time-space tradeoffs in theoretical computer science. Empirically, we show architectures that maintain edge embeddings almost always improve on their node-based counterparts—frequently significantly so in topologies that have "hub" nodes.

## 1. Introduction

Graph neural networks (GNNs) have emerged as the dominant approach for solving machine learning tasks on graphs. Over the span of the last decade, many different architectures have been proposed, both in order to improve different notions of efficiency, and to improve performance on a variety of benchmarks. Nevertheless, theoretical and empirical understanding of the impact of different architectural design choices remains elusive.

One previous line of work (Xu et al., 2019) has focused on characterizing the representational limitations stemming from the *symmetry-preserving* properties of GNNs when the node features are not informative (also called "anonymous GNNs") — in particular, relating GNNs to the Weisfeiler-Lehman graph isomorphism test (Leman & Weisfeiler, 1968). Another line of work (Oono & Suzuki, 2020) focuses on the potential pitfalls of the *(over)smoothing effect* of deep GNN architectures, with particular choices of weights and non-linearities, in an effort to explain the difficulties of training deep GNN models. Yet another (Black et al., 2023) focuses on training difficulties akin to vanishing introduced by *"bottlenecks"* in the graph topology.

In this paper, we focus on the benefits of maintaining and updating *edge embeddings* over the course of the computation of the GNN. More concretely, a typical GNN maintains a *node embedding* $h_v$ at each node $v$ of the underlying graph. In each layer of the GNN, the embedding at node $v$ is updated based on the embeddings at its neighbors. But an alternative paradigm is to maintain data at each *edge* $e$ of the graph, and to update this edge embedding based on the embeddings of the edges that share a node with $e$.

Intuitively, this paradigm seems at least as expressive as maintaining node embeddings, since in principle each edge could maintain the embeddings of its incident nodes. Additionally, there may be tasks where initial features are most naturally associated with edges (e.g., attributes of the relationship between two nodes) — or the final predictions of the network are most naturally associated with edges (e.g., in link prediction, where we want to decide which potential links are true links).

GNNs that fall in the general edge-based paradigm have been used for various applications – including link prediction (Cai et al., 2021; Liang et al., 2025) as well as reasoning about relations between objects (Battaglia et al., 2016), molecular property prediction (Gilmer et al., 2017; Choudhary & DeCost, 2021), and detecting clusters of communities in graphs (Chen et al., 2019) – with robust empirical benefits. These approaches instantiate the edge-based paradigm in a plethora of ways. However, it is difficult to disentangle to what degree performance improvements come from added information from domain-specific initial edge embeddings, versus properties of other particular archi-

---

[1]Department of EECS, Massachusetts Institute of Technology, Cambridge, MA, USA [2]Department of Machine Learning, Carnegie Mellon University, Pittsburgh, PA, USA [3]Department of Mathematics, Duke University, Durham, NC, USA. Correspondence to: Dhruv Rohatgi <drohatgi@mit.edu>, Andrej Risteski <aristesk@andrew.cmu.edu>.

*Proceedings of the $42^{nd}$ International Conference on Machine Learning*, Vancouver, Canada. PMLR 267, 2025. Copyright 2025 by the author(s).

tectural choices, versus inherent benefits of the edge-based paradigm itself (whether representational, or via improved training dynamics).

We focus on theoretically and empirically quantifying the added *representational* benefit from maintaining edge embeddings. We find that **(1)** *theoretically*, for certain graph topologies, edge embeddings can have substantial *representational* benefits in terms of the depth of the model; and **(2)** these benefits can be witnessed *empirically*, even when equalizing for all other architectural and data design choices.

## 2. Overview of results

### 2.1. Representational benefits from maintaining edge embeddings.

Our theoretical results elucidate the representational benefits of maintaining edge embeddings. More precisely, we show that there are natural tasks on graphs that can be solved by a *shallow* model maintaining constant-size edge embeddings, but can only be solved by a model maintaining constant-size node embeddings if it is much *deeper*.

To reason about the impact of depth on the representational power of edge-embedding-based and node-embedding-based architectures, we introduce two *local computation models*. In the node-embedding case (Definition 1), we assume each node of the graph $G$ supports a processor that maintains a state with a *fixed amount of memory*. In one round of computation, each node receives messages from the adjacent nodes, which are aggregated by the node into a new state. In this abstraction, we think of the memory of the processor as the total bits of information each embedding can retain, and we think of one round of the protocol as corresponding to one layer of a GNN. The edge-embedding case is formalized in a similar fashion, except that the processors are placed on the edges of the graph, and two edge processors are "adjacent" if the edges share a vertex in common (Definition 2). In both cases, the input is distributed across the edges of the graph, and is only locally accessible.

With this setup in mind, our first result focuses on *probabilistic inference* on graphs, specifically, the task of maximum a-posteriori (MAP) estimation in a pairwise graphical model on a graph $G = (V, E)$. For this task, given edge attributes describing the pairwise interactions $\phi_{\{a,b\}}$, the goal is to compute $\arg\max_{x \in \{0,1\}^V} p_\phi(x)$, where $p_\phi(x) \propto \exp\left(\sum_{\{a,b\} \in E} \phi_{\{a,b\}}(x_a, x_b)\right)$.

**A depth separation between node and edge architectures.** We show that there is a graph with $O(n)$ vertices and edges, for which MAP estimation with any node message-passing protocol requires $\Omega(\sqrt{n})$ rounds, but MAP estimation with an edge message-passing protocol only requires $O(1)$ rounds, when both protocols are restricted to $O(1)$

memory per processor. The lower bound on node message-passing protocols tracks the "flow of information" in the graph, reminiscent of graph pebbling techniques used to prove time-space tradeoffs in theoretical computer science (Grigor'ev, 1976; Abrahamson, 1991). See Theorem 1. The crucial Lemma 2 can be viewed as an information-theoretic formalization of the celebrated phenomenon of oversquashing (Alon & Yahav, 2021).

**The view from symmetry.** Above, we are not imposing any *symmetry constraints* – that is, invariance of the computation at a node or edge to its identity and the identities of its neighbors. Indeed, the edge message-passing protocol constructed above is highly non-symmetric. However, we show there is a (different, but also natural) task where *even symmetric* edge message-passing protocols achieve a better depth/memory tradeoff than node message-passing protocols. See Theorem 4.

**Importance of the memory lens.** The memory constraints are crucial for the results above. Without memory constraints, we can show that the node message-passing architecture can simulate the edge message-passing architecture, while only increasing the depth by 1 (Proposition 3). Moreover, the *symmetric* node message-passing architecture can simulate the *symmetric* edge message-passing architecture, again while only increasing the depth by 1. See Theorem 5. We view this as evidence that many fine-grained properties of architectural design for GNNs cannot be adjudicated by solely considering them through the classical lens of invariance (Xu et al., 2019).

### 2.2. Empirical benefits of edge-based architectures.

The theory, while only characterizing representational power, suggests that architectures that maintain edge embeddings should have strictly better performance compared to their node embedding counterparts. We test this in both real-life benchmarks and natural synthetic sandboxes.

**Ablation on GNN benchmarks.** We consider several popular GNN benchmarks. Equalizing for all other aspects of the architecture (e.g., depth, dimensionality of the embeddings), we find that the accuracy achieved by edge-based architectures is always comparable and typically *slightly* better than that of their node-based counterparts. This confirms that — all else being equal — the representational advantages of edge-based architectures do not introduce additional training difficulties. However, it suggests that standard benchmarks could be too easy to clearly identify better architectures.[1] Details are included in Section 8.1.

---

[1] Note, the goal of these experiments is *not* to propose a new architecture — there are already a variety of (very computationally efficient) GNNs that in some manner maintain edge embeddings.

**Synthetic stress-tests.** Next, we consider two synthetic settings to stress test the performance of edge-based architectures. Inspired by the graph topology that provides a theoretical separation between edge and node-based protocols (Theorem 1 and Theorem 4), we consider graphs in which there is a hub node, and tasks that are "naturally" solved by an edge-based architecture. Precisely, we consider a star graph, in which the labels on the leaves are generated by a "planted" edge-based architecture with randomly chosen weights. The node-based architecture, on the other hand, has to pass messages between the leaves indirectly through the center of the star. Empirically, we indeed observe that the performance of edge-based architectures is significantly better. Details are included in Section 8.2

Finally, again inspired by the theoretical setting in Theorem 1, we consider probabilistic inference on *tree graphs* — precisely, learning a GNN that calculates node marginals for an Ising model, a pairwise graphical model in which the pairwise interactions are just the product of the end points. An added motivation for this setting is the fact that belief propagation — a natural algorithm to calculate the marginals — can be written as an edge-based message-passing algorithm. Again, empirically we see robust benefits of edge-based architectures over node-based architectures, though we find that *directionality* is a more critical architecture choice. Details are included in Section 8.3.

## 3. Related Works

**The symmetry lens on GNNs:** The most extensive theoretical work on GNNs has concerned itself with the representational power of different GNN architectures, while trying to preserve equivariance (to permuting the neighbors) of each layer. (Xu et al., 2019) connected the expressive power of such architectures to the Weisfeiler-Lehman (WL) test for graph isomorphism. Subsequent works (Maron et al., 2019; Zhao et al., 2022) focused on strengthening the representational power of the standard GNN architectures from the perspective of symmetries—more precisely, to simulate the $k$-WL test, which for $k$ as large as the size of the graph becomes as powerful as testing graph isomorphism. Our work suggests this perspective is insufficient to fully understand the representational power of different architectures.

**GNNs as a computational machine:** Two recent papers (Loukas, 2020a;b) considered properties of GNNs when viewed as "local computation" machines, in which a layer of computation allows a node to aggregate the current values of the neighbors (in an arbitrary fashion, without necessarily considering symmetries). Using reductions from the CONGEST model, they provide lower bounds on width and depth for the standard node-embedding based architecture. However, they do not consider architectures with edge embeddings, which is a focus of our work.

**Communication complexity methods to prove representational separations:** Tools from distributed computation and communication complexity have recently been applied not only to understand the representational power of GNNs (Loukas, 2020a;b), but also the representational power of other architectures like transformers (Sanford et al., 2024b;a). In particular, (Sanford et al., 2024a) draws a connection between number of rounds for a MPC (Massively Parallel Computation) protocol, and the depth of attention-based architectures.

**GNNs for inference and graphical models:** (Xu & Zou, 2023) consider the approximation power of GNNs for calculating marginals for pairwise graphical models, if the family of potentials satisfies strong symmetry constraints. They do not consider the role of edge embeddings or memory.

## 4. Setup

We denote the graph associated with the GNN as $G = (V, E)$, denoting the vertex set as $V$ and the edge set as $E$. The graph induces adjacency relations on both edges and nodes. For $v, v' \in V$ and $e, e' \in E$, we have: $v \sim v'$ if $\{v, v'\} \in E$; $v \sim e$ if $e = \{u, v\}$ for some $u \in V$; and $e \sim e'$ if $e, e'$ share at least one vertex. For all graphs considered in this paper, we assume that $\{v, v\} \in E$ for all $v \in V$, so that adjacency is reflexive. We then define adjacency functions $\mathcal{N} = \mathcal{N}_G : V \cup E \to V$ and $\mathcal{M} = \mathcal{M}_G : V \cup E \to E$ as $\mathcal{N}_G(a) := \{v \in V : a \sim v\}$ and $\mathcal{M}_G(a) := \{e \in E : a \sim e\}$.[2]

**Graph Neural Networks.** A typical way to parametrize a layer $l$ of a GNN (Xu et al., 2019) is to maintain, for each node $v$ in the graph, a node embedding $h_v^{(l)}$, which is calculated in terms of its neighbor set $\mathcal{N}(v)$ as

$$a_v^{(l+1)} = \text{AGGREGATE}\Big(h_u^{(l)} : u \in \mathcal{N}(v)\Big)$$
$$h_v^{(l+1)} = \text{COMBINE}\Big(a_v^{(l+1)}, h_v^{(l)}\Big), \qquad (1)$$

for parametrized functions AGGREGATE and COMBINE. These updates can be viewed as implementing a (trained) message-passing algorithm, in which nodes pass messages to their neighbors, which are then aggregated and combined with the current state (i.e., embedding) of a node. The initial node embeddings $h_v^{(0)}$ are frequently part of the task specification (e.g., a vector of fixed features that can be associated with each node). When this is not the case, they can be set to fixed values (e.g., the all-ones vector) or random values.

But an alternative way to parametrize a layer of computation is to maintain, for each *edge* $e$, an edge embedding $h_e^{(l)}$

---

[2]The graph is assumed to be undirected, as is most common in the GNN literature. Dependence of the adjacency functions on $G$ is omitted when clear from context.

which is calculated as:

$$a_e^{(l+1)} = \text{AGGREGATE}\Big(h_a^{(l)} : a \in \mathcal{M}(e)\Big)$$

$$h_e^{(l+1)} = \text{COMBINE}\Big(a_e^{(l+1)}, h_e^{(l)}\Big). \tag{2}$$

Recall that $\mathcal{M}(e)$ denotes the "neighborhood" of edge $e$, i.e. all edges $a$ that share a vertex with $e$.

**Local memory-constrained computation.** In order to reason about the required depth with different architectures, we will define a mathematical abstraction for one layer of computation in the GNN. We will define two models for local computation, one for each of the edge-embedding and node-embedding architecture. Unlike much prior work on GNNs and distributed computation, we will also have *memory* constraints — more precisely, we will constrain the bit complexity of the node and edge embeddings being maintained.

In both models, there is an underlying graph $G = (V, E)$, and the goal is to compute a function $g : \Phi^E \to \{0,1\}^V$, where $\Phi$ is the fixed-size *input alphabet*, via several rounds of message-passing on the graph $G$. This domain of $g$ is $\Phi^E$ because in *both* models, the inputs are given on the edges of the graph — the node model will just be unable to store any *additional* information on the edges. As we will see in Section 5, this is a natural setup for probabilistic inference on graphs.

In both models, a protocol is parametrized by the number of rounds $T$ required, and the amount of memory $B$ required per local processor. For notational convenience, for $B \in \mathbb{N}$ we define $\mathcal{X}_B := \{0,1\}^B$, i.e. the length-$B$ binary strings. Recall that $\mathcal{N}(v), \mathcal{M}(v)$ denote the sets of vertices and edges adjacent to vertex $v$ in graph $G$, respectively.

**Definition 1** (Node message-passing protocol). Let $T, B \in \mathbb{N}$ and let $G = (V, E)$ be a graph. A *node message-passing protocol* $P$ on graph $G$ with $T$ rounds and $B$ bits of memory is a collection of functions $(f_{t,v})_{t \in [T], v \in V}$ where $f_{t,v} : \mathcal{X}_B^{\mathcal{N}(v)} \times \Phi^{\mathcal{M}(v)} \to \mathcal{X}_B$ for all $t, v$. For an *input* $I \in \Phi^E$, the *computation* of $P$ at a round $t \in [T]$ is the map $P_t(\cdot; I) : V \to \mathcal{X}_B$ defined inductively by $P_t(v; I) := f_{t,v}((P_{t-1}(v'; I))_{v' \in \mathcal{N}(v)}, (I(e))_{e \in \mathcal{M}(v)})$ where $P_0 \equiv 0$. We say that $P$ *computes* a function $g : \Phi^E \to \{0,1\}^V$ on inputs $\mathcal{I} \subseteq \Phi^E$ if $P_T(v; I)_1 = g(I)_v$ for all $v \in V$ and all $I \in \mathcal{I}$.

In words, the value computed by vertex $v$ at round $t$ is some function of the previous values stored at the neighbors $v' \in \mathcal{N}(v)$, as well as possibly the problem inputs on the edges adjacent to $v$ (i.e. $(I(e))_{e \in \mathcal{M}(v)}$). Note that $P_t(v; I)$ may indeed depend on $P_{t-1}(v; I)$, due to our convention that $v \in \mathcal{N}(v)$. We can define the edge message-passing protocol analogously:

**Definition 2** (Edge message-passing protocol). Let $T, B \in \mathbb{N}$ and let $G = (V, E)$ be a graph. An *edge message-passing protocol* $P$ on graph $G$ with $T$ rounds and $B$ bits of memory is a collection of functions $(f_{t,e})_{t \in [T], e \in E}$ where $f_{t,e} : \mathcal{X}_B^{\mathcal{M}(e)} \times \Phi \to \mathcal{X}_B$ for all $t, e$, together with a collection of functions $(\tilde{f}_v)_{v \in [V]}$ where $\tilde{f}_v : \mathcal{X}_B^{\mathcal{M}(v)} \to \{0,1\}$. For an *input* $I \in \Phi^E$, the *computation* of $P$ at a timestep $t \in [T]$ is the map $P_t(\cdot; I) : E \to \mathcal{X}_B$ defined inductively by: $P_t(e; I) := f_{t,e}((P_{t-1}(e'; I))_{e' \in \mathcal{M}(e)}, I(e))$ where $P_0 \equiv 0$. We say that $P$ *computes* a function $g : \Phi^E \to \{0,1\}^V$ on inputs $\mathcal{I} \subseteq \Phi^E$ if $\tilde{f}_v((P_T(e; I))_{e \in \mathcal{M}(v)}) = g(I)_v$ for all $v \in V$ and all $I \in \mathcal{I}$.

**Remark 3** (Relation to distributed computation literature). These models are very related to classical models in distributed computation like LOCAL (Linial, 1992) and CONGEST (Peleg, 2000). However, the latter models ignore memory constraints, so we cannot usefully port lower and upper bounds from this literature.

**Remark 4** (Computational efficiency). In the definitions above, we allow the update rules $f_{t,v}, f_{t,e}$ to be arbitrary functions. In particular, a priori they may not be efficiently computable. However, our results showing a function can be implemented by an edge message-passing protocol (Theorem 1, Part 2 and Theorem 4, Part 2) in fact use simple functions (computable in linear time in the size of the neighborhood), implying the protocol can be implemented in parallel (with one processor per node/edge respectively) with parallel time complexity $O(TB \cdot \max_v |\mathcal{M}(v)|)$. On the other hand, for the results showing a function cannot be implemented by a node message-passing protocol (Theorem 1, Part 1 and Theorem 4, Part 1), we prove an impossibility result for a *stronger* model (one in which the computational complexity of $f_{t,v}$ is unrestricted) — which makes our results only *stronger*.

**Symmetry-constrained protocols.** Typically, GNNs are architecturally constrained to respect the symmetries of the underlying graph. Below we formalize the most natural notion of symmetry in our models of computation. Note, our abstraction of a round in the message-passing protocol generalizes the notion of a layer in a graph neural network—and the abstraction defined below correspondingly generalizes the standard definition of permutation equivariance (Xu et al., 2019). We use the notation $\{\!\{\}\!\}$ to denote a multiset.

**Definition 5** (Symmetric node message-passing protocol). A node message-passing protocol $P = (f_{t,v})_{t \in [T], v \in V}$ on graph $G = (V, E)$ is *symmetric* if there are functions $(f_t^{\mathsf{sym}})_{t \in [T]}$ so that for every $t \in [T]$ and $v \in V$, the function $f_{t,v}$ can be written as:

$$f_{t,v}((c(v'))_{v' \in \mathcal{N}(v)}, (I(e))_{e \in \mathcal{M}(v)})$$
$$= f_t^{\mathsf{sym}}(c(v), \{\!\{(c(v'), I(\{v, v'\})) : v' \in \mathcal{N}(v)\}\!\}).$$

Our definition of a symmetric edge message-passing protocol is analogous; we defer it to Appendix A due to space constraints.

**Additional notation.** For a set $K$ contained in universe $U$, we let $\overline{K} := K \setminus U$ denote the complement of $K$ in $U$ (where $U$ will be clear from context). We let $\Delta(S)$ denote the family of distributions over set $S$.

## 5. Depth separation between edge and node message passing protocols under memory constraints

We now consider a common task in probabilistic inference on a *pairwise graphical model*: calculating the MAP (maximum a-posterior) configuration.

**Definition 6** (Pairwise graphical model)**.** For any graph $G = (V, E)$, the *pairwise graphical model* on $G$ with potential functions $\phi_{\{a,b\}} : \{0,1\}^2 \to \mathbb{R}$ is the distribution $p_\phi \in \Delta(\{0,1\}^V)$ defined as $p_\phi(x) \propto \exp\left(-\sum_{\{a,b\}\in E} \phi_{\{a,b\}}(x_a, x_b)\right)$.

**Definition 7** (MAP evaluation)**.** Let $\Phi \subseteq \{\phi : \{0,1\}^2 \to \mathbb{R}\}$ be a finite set of potential functions. A *MAP (maximum a-posteriori) evaluator for $G$* (with potential function class $\Phi$) is any function $g : \Phi^E \to \{0,1\}^V$ that satisfies $g(\phi) \in \arg\max_{x \in \{0,1\}^V} p_\phi(x)$ for all $\phi \in \Phi^E$.

With this setup in mind, we will show that there exists a pairwise graphical model, and a local function class $\Phi$, such that an edge message passing protocol can implement MAP evaluation with a constant number of rounds and a constant amount of memory, while any node message protocol with $T$ rounds and $B$ bits of memory requires $TB = \Omega(\sqrt{|V|})$. Precisely, we show:

**Theorem 1** (Main, separation between node and edge message-passing protocols)**.** *Fix $n \in \mathbb{N}$. There is a graph $G$ with $O(n)$ vertices and $O(n)$ edges, and a function class $\Phi$ of size $O(1)$, so that:*

1. *Let $g$ be any MAP evaluator for $G$ with potential function class $\Phi$. Any node message-passing protocol on $G$ with $T$ rounds and $B$ bits of memory that computes $g$ requires $TB \geq \sqrt{n} - 1$.*

2. *There is an edge message-passing protocol $(f_{t,e})_{t,e}$ on $G$ with $O(1)$ rounds and $O(1)$ bits of memory that computes a MAP evaluator for $G$ with potential function class $\Phi$. Additionally, for all $t, e$, the update rule $f_{t,e}$ can be evaluated in $O(|\mathcal{M}(e)|)$ time.*

The construction for the above result also implies a separation between the parallel time complexity of node and edge message-passing protocols (with one processor per node or

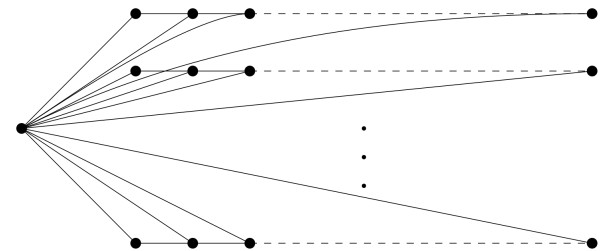

*Figure 1.* The graph $G$ for which Theorem 1 exhibits a separation between edge message-passing and node message-passing. The graph consists of $\sqrt{n}$ paths of length $\sqrt{n}$, as well as a single "hub vertex" connected to all other vertices.

edge, respectively), assuming that each processor reads its entire input in each round. The graph $G$ has a node with $\Omega(n)$ neighbors, so in any node message-passing protocol its processor must perform $\Omega(nB)$ computation per round, for overall time complexity of $\Omega(nBT) = \Omega(n^{3/2})$. In contrast, the constructed edge message-passing protocol is implementable in time $O(n)$ per round, and hence $O(n)$ time overall.

We provide a proof sketch of the main techniques here, and relegate the full proofs to Appendix B. The graph $G$ that exhibits the claimed separation is a disjoint union of $\sqrt{n}$ path graphs, with an additional "hub vertex" that is connected to all other vertices in the graph (Fig. 1). The intuition for the separation is that MAP estimation requires information to disseminate from one end of each path to the other, and the hub node is a bottleneck for node message-passing but not edge message-passing. We expand upon both aspects of this intuition below.

**Lower bound for node message-passing protocols:** Our main technical lemma for the first half of the theorem is Lemma 2. It gives a generic framework for lower bounding the complexity of any node message-passing protocol that computes some function $g$, by exhibiting a set of nodes $S \subset V$ where computing $g$ requires large "information flow" from distant nodes. More precisely, for any fixed set of "bottleneck nodes" $K$, consider the radius-$T$ neighborhood of $S$ when $K$ is removed from the graph. In any $T$-round protocol, input data from outside this neighborhood can only reach $S$ by passing through $K$. But the total number of bits of information computed by $K$ throughout the protocol is only $TB|K|$. This gives a bound on the number of values achievable by $g$ on $S$. We formalize this argument below (proof in Appendix B):

**Lemma 2.** *Let $G = (V, E)$ be a graph. Let $P$ be a node message-passing protocol on $G$ with $T$ rounds and $B$ bits of memory, which computes a function $g : \Phi^E \to \{0,1\}^V$. Pick any disjoint sets $K, S \subseteq V$. Define $H := G[\bar{K}], F := \mathcal{M}(\mathcal{N}_H^{T-1}(S))$, where $\mathcal{N}_H^{T-1}(S)$ is the $(T-1)$-hop neigh-*

*borhood of S in H.*

*Then:* $TB \geq \frac{1}{|K|} \log \max_{I_F \in \Phi^F} \left| \left\{ g_S \left( I_F, I_{\overline{F}} \right) : I_{\overline{F}} \in \Phi^{\overline{F}} \right\} \right|.$

**Remark 8.** The proof technique is inspired by and related to classic techniques (specifically, Grigoriev's method) for proving time-space tradeoffs for restricted models of computation like branching programs ((Grigor'ev, 1976), see Chapter 10 in (Savage, 1998) for a survey). There, one defines the "flow" of a function, which quantifies the existence of subsets of coordinates, such that setting them to some value, and varying the remaining variables results in many possible outputs. In our case, the choice of subsets is inherently tied to the topology of the graph $G$. Our technique is also inspired by and closely related to the "light cone" technique for proving round lower bounds in the LOCAL computation model (Linial, 1992). However, our technique takes advantage of bottlenecks in the graph to prove stronger lower bounds (which would be impossible in the LOCAL model where memory constraints are ignored).

**Remark 9** (Relation to oversquashing and virtual nodes)**.** The above lemma can be seen as an information-theoretic formalization and explanation of the commonly observed phenomenon of "oversquashing" (Alon & Yahav, 2021), wherein GNNs on graphs with a hub node that needs to "pass information" between two large sets of nodes seem to perform poorly. It also demonstrates a difficulty for memory-constrained virtual nodes — which are hub nodes that are often artificially added to graphs in practice (Cai et al., 2023).

The proof of Part 1 of Theorem 1 now follows from an application of Lemma 2 with a particular choice of $K$ and $S$. Specifically, we choose $K$ to be the "hub" node (i.e. $K = \{0\}$) and $S$ to be the set of left endpoints of each path. To show that any MAP evaluator has large information flow to $S$ (in the quantitative sense of Lemma 2), it suffices to observe that in a pairwise graphical model on $G$ where a different external field is applied to the right endpoint of each path, and all pairwise interactions along paths are positive, the MAP estimate on each vertex in $S$ must match the external field on the corresponding right endpoint.

**Upper bound for edge message-passing protocols:** The key observation for constructing a constant-round edge message-passing protocol for MAP estimation on $G$ is that all of the input data can be collected on the edges adjacent to the hub vertex. At this point, every such edge has access to all of the input data, and hence can evaluate the function. If $G$ were an arbitrary graph, this final step would potentially be NP-hard. However, since the induced subgraph after removing the hub vertex is a disjoint union of paths, in fact there is a linear-time dynamic programming algorithm for MAP estimation on $G$ (Lemma 6). This completes the proof overview for Theorem 1.

The separation discussed above crucially relies on the existence of a high-degree vertex in $G$. When the maximum degree of $G$ is bounded by some parameter $\Delta$, it turns out that any edge message-passing protocol can be simulated by a node message-passing protocol with roughly the same number of rounds and only a $\Delta$ factor more memory per processor. The idea is for each node to simulate the computation that would have been performed (in the edge message-passing protocol) on the adjacent edges. The following proposition formalizes this idea (proof in Appendix B):

**Proposition 3.** *Let $T, B \geq 1$. Let $G = (V, E)$ be a graph with maximum degree $D$. Let $P$ be an edge message-passing protocol on $G$ with $T$ rounds and $B$ bits of memory. Then there is a node message-passing protocol $P'$ on $G$ that computes $P$ with $T + 1$ rounds and $O(DB)$ bits of memory.*

## 6. Depth separation under memory and symmetry constraints

One drawback of the separation in the previous section is that the constructed edge protocol was highly non-symmetric, whereas in practice GNN protocols are typically architecturally constrained to respect the symmetries of the underlying graph. In this section we prove that there is a separation between the memory/round trade-offs for node and edge message-passing protocols even under additional symmetry constraints.

**Theorem 4.** *Let $n \in \mathbb{N}$. There is a graph $G = (V, E)$ with $O(n)$ vertices and $O(n)$ edges, and a function $g : \{0, 1\}^E \to \{0, 1\}^V$, so that:*

1. *Any node message-passing protocol on $G$ with $T$ rounds and $B$ bits of memory that computes $g$ requires $TB \geq \Omega(\sqrt{n})$.*

2. *There is a symmetric edge message-passing protocol on $G$ with $O(1)$ rounds and $O(\log n)$ bits of memory that computes $g$.*

For intuition, we sketch the proof of a relaxed version of the theorem where the input alphabet is $[n]$. It is conceptually straightforward to adapt the construction to binary alphabet (essentially, by adding new vertices and using a unary encoding). We defer the full proof to Appendix C.

Let $G = (V, E)$ be a star graph with root node 0 and leaves $\{1, \ldots, n\}$. We define a function $g : [n]^E \to \{0, 1\}^V$ by $g(I)_v = 1$ if and only if there is some edge $e \neq \{0, v\}$ such that $I(e) = I(\{0, v\})$, i.e. the input on edge $\{0, v\}$ equals the input on some other edge. Since $g$ is defined to be equivariant to relabelling the edges, and all edges are incident to each other, it is straightforward to see that there is a symmetric one-round edge message-passing protocol that computes $g$ with $O(\log n)$ memory (in contrast, the edge

message-passing protocol constructed in Section 5 was not symmetric, as it required that the edges incident to the high-degree vertex were labelled by which path they belonged to). However, there is no low-memory, low-round *node* message-passing algorithm. Informally, this is because vertex 0 is an information bottleneck, and $\Omega(n)$ bits of information need to pass through it. Similar to in Section 5, this intuition can be made formal using Lemma 2.

# 7. Symmetry alone provides no separation

In the previous sections we saw that examining *memory constraints* yields a separation between different GNN architectures (whether or not we take symmetry into consideration). In this section, we consider what happens if we solely consider *symmetry constraints* (that is, constraints imposed by requiring that the computation in a round of the protocol is invariant to permutations of the order of the neighbors). This viewpoint was initiated by (Xu et al., 2019), who showed that when the initial node features are uninformative (that is, the same for each node), a standard GNN necessarily outputs the same answer for two graphs that are 1-Weisfeiler-Lehman equivalent (that is, graphs that cannot be distinguished by the Weisfeiler-Lehman test, even though they may not be isomorphic).

To be precise, we revisit the representational power of symmetric GNN architectures in the setting where the input features may be distinct and informative. We show that *if we remove the memory constraints* from Section 5, but *impose permutation invariance* for the computation in each round, any function that is computable by a $T$-layer edge message-passing protocol can be computed by a $(T + 1)$-layer node message-passing protocol. Note that this statement is incomparable to Proposition 3 because we impose constraints on symmetry, but remove constraints on memory.

**Theorem 5** (No separation under symmetry constraints). *Let $T \geq 1$. Let $P$ be a symmetric edge message-passing protocol (Definition 11) on graph $G = (V, E)$ with $T$ rounds. Then there is a $(T + 1)$-round symmetric node message-passing protocol (Definition 5) $P'$ on $G$ that computes the same function as $P$.*

**Remark 10.** Theorem 5 and its proof are closely related to the fact that the 1-Weisfeiler-Lehman test is equivalent to the 2-Weisfeiler-Lehman test, which was reintroduced in the context of higher-order GNNs (Huang & Villar, 2021). The main difference is conceptual rather than technical. Prior works on expressivity of GNNs (with respect to the Weisfeiler-Lehman test) measure expressivity by asking "for a given input, what are the possible outputs" (Xu et al., 2019). In contrast, particularly for computation on graphs with informative input labels, it is natural to ask what *functions* a GNN can represent, i.e. "what are the possible mappings from inputs to outputs". In the above result (and

throughout the paper) we take this functional perspective, analogous to the classical representational theory for neural networks (Telgarsky, 2016). Theorem 5 shows that even with arbitrary input features on the edges, the computation of a GNN with edge embeddings and symmetric updates can be simulated by a GNN with only node embeddings, without losing symmetry.

To prove Theorem 5, note that it suffices to simulate the protocol $P$ for which the update rules $f^{\text{sym}}, \tilde{f}^{\text{sym}}$ in Definition 11 are identity functions on the appropriate domains. In order to simulate $P$, we construct a symmetric node message-passing protocol $P'$ for which the computation at time $t + 1$ and node $v$ on input $I$ is the multiset of features computed by $P$ at time $t$ at edges adjacent to $v$: $Q_t(v; I) := \{\!\!\{ P_t(e; I) : e \in \mathcal{M}(v) \}\!\!\}$. This is possible since the computation of $P$ at time $t$ and edge $e = (u, v)$ is $P_t(e; I) = (I(e), P_{t-1}(e; I), \{\!\!\{ Q_{t-1}(u; I), Q_{t-1}(v; I) \}\!\!\})$. The node message-passing protocol is tracking $Q_{t-1}(\cdot; I)$; moreover, it can recursively compute $P_{t-1}(e; I)$ using the same formula. See Appendix D for the formal proof.

# 8. Empirical benefits of edge-based architectures

In this section we demonstrate that the representational advantages the theory suggests are borne out by experimental evaluations, both on real-life benchmarks and two natural synthetic tasks we provide. Note that all the experiments were done on a machine with 8 Nvidia A6000 GPUs.

### 8.1. Performance on common benchmarks

First we compare the performance of the most basic GNN architecture (Graph Convolutional Network, (Kipf & Welling, 2017)) with node versus edge embeddings. In the notation of (1) and (2), the AGGREGATE and COMBINE operations are integrated together, giving either (3) or (4):[3]

$$h_v^{(l+1)} = h_v^{(l)} + \sigma\big(W^{(l)} \text{AVG}\big(h_w^{(l)} : w \in \mathcal{N}(v) \setminus \{v\}\big)\big) \ (3)$$

$$h_e^{(l+1)} = h_e^{(l)} + \sigma\big(W^{(l)} \text{AVG}\big(h_f^{(l)} : f \in \mathcal{M}(e) \setminus \{e\}\big)\big) \ (4)$$

for trained matrices $W^{(l)}$ and a choice of non-linearity $\sigma$. The only difference between these architectures is that in the latter case, the message passing happens over the *line graph* of the original graph (i.e. the neighborhood of an edge is given by the other edges that share a vertex with it) — thus, this can be viewed as an ablation experiment in which the only salient difference is the type of embeddings being maintained. To also equalize the information in the input embeddings, we only use the node embeddings in the benchmarks we consider: for the edge-based architecture

---

[3]This is the "residual" parametrization, which we use in experiments unless otherwise stated.

*Table 1.* Comparison of node-based (3) and edge-based (4) GCN architectures across various graph benchmarks. The performance of the edge-based architecture robustly matches or improves the node-based architecture.

| Model | ZINC | MNIST | CIFAR-10 | Peptides-Func | Peptides-Struct |
|---|---|---|---|---|---|
| | MAE ($\downarrow$) | ACCURACY ($\uparrow$) | ACCURACY ($\uparrow$) | AP ($\uparrow$) | MAE ($\downarrow$) |
| GCN | $0.3430 \pm 0.034$ | $\mathbf{95.29 \pm 0.163}$ | $55.71 \pm 0.381$ | $0.6816 \pm 0.007$ | $0.2453 \pm 0.0001$ |
| Edge-GCN (Ours) | $\mathbf{0.3297 \pm 0.011}$ | $94.37 \pm 0.065$ | $\mathbf{57.44 \pm 0.387}$ | $\mathbf{0.6867 \pm 0.004}$ | $\mathbf{0.2437 \pm 0.0005}$ |

(2), we initialize the edge embeddings by the concatenation of the node embeddings of the endpoints.

In Table 1, we show that *this single change* (without any other architectural modifications) uniformly results in the edge-based architecture at least matching the performance of the node-based architecture, sometimes improving upon it. *Note, the purpose of this table is not to advocate a new GNN architecture*[4]— but to confirm that the increased representational power of the edge-based architecture indicated by the theory also translates to improved performance when the model is trained. For each benchmark, we follow the best performing training configuration as delineated in (Dwivedi et al., 2023).

### 8.2. A synthetic task for topologies with node bottlenecks

The topologies of the graphs in Theorem 1 and Theorem 4 both involve a "hub" node, which is connected to all other nodes in the graph. Intuitively, in node-embedding architectures, such nodes have to mediate messages between many pairs of other nodes, which is difficult when the node is constrained by memory. To empirically stress test this intuition, we produce a synthetic dataset and train a GNN to solve a regression task on a graph with a fixed *star-graph* topology— a simpler topology than the constructions in Theorem 1 and Theorem 4—but capturing the core aspect of both. A star graph is a graph with a center node $v_0$, a set of $n$ leaf nodes $\{v_i\}_{i \in [n]}$, and edge set $\{\{v_0, v_i\}_{i \in [n]}\}$. A training point in the dataset is a list $(x_i, y_i)_{i=1}^{n}$ where $x_i$ is the *input feature* and $y_i$ is the *label* for leaf node $v_i$.

The input features are in $\mathbb{R}^{10}$, and sampled from a standard Gaussian. The labels $y_i$ are produced as outputs of a *planted* edge-based architecture. Namely, for a standard edge-based GCN as in (4), we randomly choose values for the matrices $\{W_i\}_{i \in [k]}$ for some number of layers $k$, and set the labels to be the output of this edge-based GCN, when the initial edge features to the GCN are set as $h_{\{v_0, v_i\}}^{(0)} := x_i$, i.e. the input feature $x_i$ at the corresponding leaf $i$. In Table 2, we show the performance of edge-based and node-based architectures on this dataset, varying the number of leaves $n$

---

[4]In particular, the edge-based architecture is often much more computationally costly to evaluate.

in the star graph and the depth $k$ of the planted edge-based model. In each case, the numbers indicate RMSE of the best-performing edge-based and node-based architecture, sweeping over depths up to 10 ($2\times$ the planted model), widths $\in \{16, 32, 64\}$, and a range of learning rates.

Since the planted edge-based model satisfies both *invariance* constraints (by design of the GCN architecture) and *memory* constraints (since the planted model maintains 10-dimensional embeddings), we view these results as empirical corroboration of Theorem 4—and even for simpler topologies than the proof construction.

### 8.3. A synthetic task for inference in Ising models

Finally, motivated by the probabilistic inference setting in Theorem 1, we consider a synthetic sandbox of using GNNs to predict the values of marginals in an Ising model (Ising, 1924; Onsager, 1944) – a natural type of pairwise graphical model where each node takes a value in $\{\pm 1\}$, and each edge potential is a weighted product of the edge endpoint values. Concretely, the probability distribution of an Ising model over graph $G = (V, E)$ has the form: $\forall x \in \{\pm 1\}^n$ :

$$p_{J,h}(x) \propto \exp\left(\sum_{\{i,j\} \in E} J_{\{i,j\}} x_i x_j + \sum_{i \in V} h_i x_i\right).$$

Similar to in Section 8.2, we construct a training set where the graph $G$ and and edge potentials stay fixed (precisely, $J_{i,j} = 1$ for all $\{i, j\} \in E$). A training data-point consists of a vector of node potentials $\{h_i\}_{i \in [n]}$, and labels $\{\mathbb{E}[x_i]\}_{i \in [n]}$ consisting of the marginals from the resulting Ising model $p_{J,h}$. The node potentials are sampled from a standard Gaussian distribution.

*Table 2.* Performance (in RMSE $\downarrow$) of edge-based and node-based architectures on a star-graph topology. The first number is the performance of the best edge-based model, and the second is the best node-based model, across a range of depths up to 10, widths $\in \{16, 32, 64\}$, and a range of learning rates.

| # of Leaves | Depth of Planted Model (RMSE) | | | | | |
|---|---|---|---|---|---|---|
| | 5 | | 3 | | 1 | |
| | Edge | Node | Edge | Node | Edge | Node |
| 64 | 0.004 | 0.379 | 0.011 | 0.360 | 0.008 | 0.375 |
| 32 | 0.003 | 0.366 | 0.005 | 0.363 | 0.003 | 0.361 |
| 16 | 0.007 | 0.334 | 0.002 | 0.210 | 0.002 | 0.285 |

There is a natural connection between GNNs and calculating marginals: a classical way to calculate $\{\mathbb{E}[x_i]\}$ when $G$ is a *tree* is to iterate a message passing algorithm called *belief propagation* (7), in which for each edge $\{i, j\}$ and direction $i \to j$, a message $\nu_{i \to j}^{(t+1)}$ is calculated that depends on messages $\{\nu_{k \to i}^{(t)}\}_{\{k,i\} \in E}$. The belief-propagation updates (7) naturally fit the general edge-message passing paradigm from (2). In fact, they fit even more closely a "directed" version of the paradigm, in which each edge $\{i, j\}$ maintains two embeddings $h_{i \to j}, h_{j \to i}$, such that the embedding for direction $h_{i \to j}$ depends on the embeddings $\{h_{k \to i}\}_{\{k,i\} \in E}$ — and it is possible to derive a similar "directed" node-based architecture (See Appendix F.2). For both the undirected and directed version of the architecture, we see that maintaining edge embeddings gives robust benefits over maintaining node embeddings—for a variety of tree topologies including complete binary trees, path graphs, and uniformly randomly sampled trees of a fixed size. Details are included in Appendix F.

## 9. Conclusions and future work

Graph neural networks are the best-performing machine learning method for many tasks over graphs. There is a wide variety of GNN architectures, which frequently make opaque design choices and whose causal influence on the final performance is difficult to understand and estimate. In this paper, we focused on understanding the impact of maintaining edge embeddings on the representational power, as well as the subtleties of considering constraints like memory and invariance. We highlight two notable directions for future work:

1. One significant downside of maintaining edge embeddings is the *computational* overhead on dense graphs. Hence, a fruitful direction for future research would be to explore more computationally efficient variants of edge-based architectures that preserve their representational power and performance. While there are a variety of interesting heuristics aimed at improving the performance of GNNs (for example, graph rewiring (Topping et al., 2022)), understanding for which graphs and tasks these heuristics can improve representational tradeoffs is largely an open question.

2. The empirical results on common graph benchmarks (Table 1) suggest much weaker benefits of edge embeddings than the synthetic (Table 2) and theoretical results. These benchmarks also largely have less skewed degree statistics. Experimenting with a wider range of benchmarks and understanding the impact of degree on performance could yield valuable additional insights.

## Acknowledgments

We thank the anonymous reviewers for their feedback that improved the presentation of this paper.

## Impact Statement

This paper presents work whose goal is to advance the field of Machine Learning. There are many potential societal consequences of our work, none which we feel must be specifically highlighted here.

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

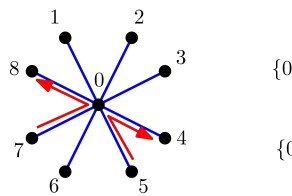
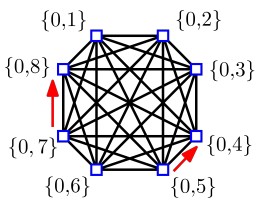

*Figure 2.* A visualization of the information bottleneck induced by "hub nodes", which is the key intuition behind Theorems 1 and 4. Here, $G$ is a star graph with $n = 8$ leaves. *Left:* The star graph $G$ itself, which describes the connectivity of the processors in a node-based message-passing protocol on $G$. Any message between two leaves must pass through the hub node (as depicted by the red arrows). Since the hub node has only constant memory, if all nodes need to pass information then intuitively $\Omega(n)$ rounds are necessary. *Right:* The line graph $L(G)$, which describes the connectivity of the processors in an edge-based message-passing protocol on $G$. Each box corresponds to an edge of the original graph. Messages can be passed directly between boxes (as depicted by the red arrows), so there is no bottleneck.

# Appendix

## A. Omitted definitions

**Definition 11** (Symmetric edge message-passing protocol). An edge message-passing protocol $P = ((f_{t,e})_{t \in [T], e \in E}, (\tilde{f}_v)_{v \in V})$ on graph $G = (V, E)$ is *symmetric* if there are functions $(f_t^{\mathsf{sym}})_{t \in [T]}$ and $\tilde{f}^{\mathsf{sym}}$ so that for every $t \in [T]$ and $e = \{u, v\} \in E$, the function $f_{t,e}$ can be written as:

$$f_{t,e}((c(e'))_{e' \in \mathcal{M}(e)}, I(e))$$
$$= f_t^{\mathsf{sym}}(I(e), c(e), \{\!\{\{\!\{c(e') : e' \in \mathcal{M}(u)\}\!\},$$
$$\{\!\{c(e') : e' \in \mathcal{M}(v)\}\!\}\}\!\}),$$

and for every $v \in V$, $\tilde{f}_v$ can be written as $\tilde{f}_v((c(e))_{e \in \mathcal{M}(v)}) = \tilde{f}^{\mathsf{sym}}(\{\!\{c(e) : e \in \mathcal{M}(v)\}\!\})$.

## B. Omitted Proofs from Section 5

In this section we give the omitted proofs from Section 5. In particular we give the formal proof of Theorem 1, which states that there is a depth separation between edge message-passing protocols and node message-passing protocols for a natural MAP estimation problem on the underlying graph $G$. Additionally, see Fig. 2 for a visualization of the key insight behind the proofs of Theorems 1 and 4: the "hub node" information bottleneck. Finally, we remark that a quantitatively stronger (and in fact tight) separation is possible if one considers general tasks rather than MAP estimation tasks – see Appendix E.

*Proof of Lemma 2.* First, we argue by induction that for each $t \in [T]$ and $v \in V \setminus K$, $P_t(v; I)$ is determined by

$I_{\mathcal{M}(N_H^{t-1}(v))}$ and $(P_\ell(k; I))_{\ell \in [t], k \in K}$. Indeed, by definition, $P_1(v; I)$ is determined by $I_{\mathcal{M}^1(v)}$ for any $v \in V \setminus K$. For any $t > 1$ and $v \in V \setminus K$, $P_t(v; I)$ is determined by $(P_{t-1}(v'; I))_{v' \in \mathcal{N}(v)}$ and $(I(e))_{e \in \mathcal{M}(v)}$. Note that $\mathcal{N}(v) \subseteq N_H(v) \cup K$. Thus, using the induction hypothesis for each $v' \in N_H(v)$, we get that $(P_{t-1}(v'; I))_{v' \in \mathcal{N}(v)}$ is determined by $\bigcup_{v' \in N_H(v)} I_{\mathcal{M}(N_H^{t-2}(v'))}$ and $(P_\ell(k; I))_{\ell \in [t], k \in K}$. So $P_t(v; I)$ is determined by $I_{\mathcal{M}(N_H^{t-1}(v))}$ and $(P_\ell(k; I))_{\ell \in [t], k \in K}$, completing the induction.

Since $P$ computes $g$ and $S \subseteq V \setminus K$, we get that $g_S(I)$ is determined by $I_{\mathcal{M}(N_H^{T-1}(S))} = I_F$ and $(P_\ell(k; I))_{\ell \in [T], k \in K}$. Thus, for any fixed $I_F \in \Phi_F$, we have

$$\left| \left\{ g_S(I_F, I_{\overline{F}}) : I_{\overline{F}} \in \Phi^{\overline{F}} \right\} \right|$$
$$\leq \left| \left\{ (P_\ell(k; (I_F, I_{\overline{F}})))_{\ell \in [T], k \in K} : I_{\overline{F}} \in \Phi^{\overline{F}} \right\} \right|$$
$$\leq |\mathcal{X}_B|^{T|K|}$$
$$= 2^{TB|K|}.$$

The lemma follows. $\qquad\square$

*Proof of Theorem 1.* Let $G$ be the graph on vertex set $V := \{0\} \cup [\sqrt{n}] \times [\sqrt{n}]$ with edge set defined below (see also Fig. 1):

$$E := \{\{0, (i, j)\} : i, j \in [\sqrt{n}]\}$$
$$\cup \{\{(i, j), (i + 1, j)\} : 2 \leq i \leq \sqrt{n}, 1 \leq j \leq \sqrt{n}\}.$$

Let $\Phi$ be the following set of four potential functions:

$$\Phi := \{(x_a, x_b) \mapsto \mathbb{1}[x_a \neq x_b],$$
$$(x_a, x_b) \mapsto \mathbb{1}[x_a \neq 1 \vee x_b \neq 1],$$
$$(x_a, x_b) \mapsto \mathbb{1}[x_a \neq 0 \vee x_b \neq 0],$$
$$(x_a, x_b) \mapsto 0\}.$$

**Lower bound.** We start by proving the lower bound against node message-passing protocols, using Lemma 2. Let $g : \Phi^E \to \{0, 1\}^V$ be any MAP evaluator for $G$ with potential function class $\Phi$, and consider any node message-passing protocol on $G$ with $T$ rounds and $B$ bits of memory that computes $g$. Let $K = \{0\}$ (the "hub node" of graph $G$) and $S = \{(1, j) : j \in [\sqrt{n}]\}$ (the set of left-hand endpoints of the paths in $G$). Suppose that $T \leq \sqrt{n} - 2$. Let $F := \mathcal{M}(N_H^{T-1}(S))$. By assumption on $T$, we have that $\{(\sqrt{n} - 1, j), (\sqrt{n}, j)\} \notin F$ for all $j \in [\sqrt{n}]$.

Let $I_F : F \to \Phi$ be the mapping that assigns the function $(x_a, x_b) \mapsto 0$ to each edge $\{0, (i, j)\} \in F$ and $(x_a, x_b) \mapsto \mathbb{1}[x_a \neq x_b]$ to each edge $\{(i, j), (i + 1, j)\} \in F$. Intuitively, this means that we are focusing on the graphical models where the hub node has no interactions with the rest of the

graph, and every non-hub-node is incentivized to match its neighbors.

We claim that even after fixing the potentials on $F$, the number of restrictions of a possible MAP assignment to the set $S$ is still exponentially large:

$$\left| \left\{ g_S(I_F, I_{\overline{F}}) : I_{\overline{F}} \in \Phi^{\overline{F}} \right\} \right| \geq 2^{\sqrt{n}}. \tag{5}$$

Indeed, for any string $y \in \{0,1\}^{\sqrt{n}}$, consider the mapping $I_{\overline{F}} : \overline{F} \to \Phi$ that assigns the function $(x_a, x_b) \mapsto \mathbb{1}[x_a \neq y_j \vee x_b \neq y_j]$ to each edge $\{(\sqrt{n}-1, j), (\sqrt{n}, j)\} \in F$, assigns $(x_a, x_b) \mapsto 0$ to each edge $\{0, (i,j)\} \in E \setminus F$, and assigns $(x_a, x_b) \mapsto \mathbb{1}[x_a \neq x_b]$ to all remaining edges in $E \setminus F$. Then every minimizer of

$$\min_{x \in \{0,1\}^V} \sum_{\{a,b\} \in E} I_{\{a,b\}}(x_a, x_b)$$

satisfies $x_{(1,j)} = \cdots = x_{(\sqrt{n},j)} = y_j$ for all $j \in [\sqrt{n}]$, since $I_{\overline{F}}$ incentivizes $x_{(\sqrt{n},j)} = y_j$ and there are positive interactions along each path. Hence, $g_S(I_F, I_{\overline{F}}) = y$. Since $y$ was chosen arbitrarily, this proves the claim (5). But now Lemma 2 implies that $TB \geq \sqrt{n}$.

We now construct an edge message-passing protocol $P$ on $G$ with $T = 3$ and $B = 4$. We (arbitrarily) identify $\Phi$ with $\{0,1\}^2$. Intuitively, the three steps of the protocol do the following:

1. First, each edge not adjacent to the hub node "reads" its own input.

2. Second, each edge $\{0, (i,j)\}$ adjacent to the hub node 0 "reads" its own input and the input of the adjacent edge $\{(i,j), (i+1,j)\}$.

3. Third, each edge $\{0, (i,j)\}$ adjacent to the hub node computes the MAP estimate of the graphical model specified by the input, using the fact that all of the data is now stored on edges incident to $\{0, (i,j)\}$. It then stores the indices of this MAP estimate corresponding to node 0 and node $(i,j)$.

We proceed to make this idea formal, which requires defining a collection of update functions $(f_{t,e})_{t,e}$. For all $i, j \in \sqrt{n}$, define update functions

$$f_{1,\{(i,j),(i+1,j)\}}(x,y) := y \qquad \text{if } i < \sqrt{n}$$
$$f_{2,\{0,(i,j)\}}(x,y) := (x_{\{(i,j),(i+1,j)\}}, x_{\{0,(i,j)\}}) \quad \text{if } i < \sqrt{n}$$
$$f_{3,\{0,(i,j)\}}(x,y) := (g_0(J(x)), g_{(i,j)}(J(x)))$$

where the second line is well-defined since edge $\{0, (i,j)\}$ is adjacent to both itself and edge $\{(i,j), (i+1,j)\}$; and in

the third line the function is computing $g_0$ and $g_{(i,j)}$ on the input $J(x) \in \Phi^E$ defined as

$$J(x)_e := \begin{cases} (x_{\{0,(k,\ell)\}})_{1:2} & \text{if } e = \{(k,\ell), (k+1,\ell)\} \\ (x_{\{0,(k,\ell)\}})_{3:4} & \text{if } e = \{0, (k,\ell)\} \end{cases},$$

where we use the notation $v_{a:b}$ for a vector $v$ and indices $a, b \in \mathbb{N}$ to denote $(v_a, v_{a+1}, \ldots, v_b)$. Note that $J(x)$ is a well-defined function of $x$ for every edge $\{0, (i,j)\}$, because $\{0, (i,j)\} \sim \{0, (k,\ell)\}$ for all $i, j, k, \ell \in [n]$. Finally, define all other functions $f_{t,e}$ to compute the all-zero function, and define

$$\tilde{f}_v(x) := \begin{cases} (x_{\{0,(1,1)\}})_{1:2} & \text{if } v = 0 \\ (x_{\{0,v\}})_{3:4} & \text{otherwise} \end{cases}.$$

This function is well-defined since $v = 0$ is adjacent to edge $\{0, (1,1)\}$ and any vertex $v \in V \setminus \{0\}$ is adjacent to edge $\{0, v\}$.

Fix any $I \in \Phi^E$. From the definition, it's clear that $P_2(\{0, (i,j)\}; I) = (I_{\{(i,j),(i+1,j)\}}, I_{\{0,(i,j)\}})$ for all $I$ and $(i,j) \in [\sqrt{n}-1] \times [\sqrt{n}]$. Hence $J((P_2(e'; I))_{e' \in \mathcal{M}(e)})_e = I$ for all edges $e$ of the form $(0, \{i,j\})$, and so $P_3(\{0, (i,j)\}; I) = (g_0(I), g_{(i,j)}(I))$ for all $(i,j) \in [\sqrt{n}] \times [\sqrt{n}]$. This means that $\tilde{f}_v((P_3(e; I))_{e \in \mathcal{M}(v)}) = g(I)_v$ for all $v \in V$, so the protocol indeed computes $g$.

It remains to argue about the computational complexity of the updates $f_{t,e}$. It's clear that for all $e \in E$ and $t \in \{1, 2\}$, the function $f_{t,e}$ can be evaluated in input-linear time. The only case that requires proof is when $t = 3$ and $e = \{0, (i,j)\}$ for some $i, j \in \sqrt{n}$. In this case $|\mathcal{M}(e)| = \Theta(n)$, so it suffices to give an algorithm for evaluating the function $g : \Phi^E \to \{0,1\}^V$ on an explicit input $J$ in $O(n)$ time. This can be accomplished via dynamic programming (Lemma 6). $\qquad \square$

**Lemma 6.** *Fix $n \in \mathbb{N}$. Let $G$, $\Phi$ be as defined in Theorem 1. Then there is an $O(n)$-time algorithm that computes a MAP evaluator for $G$ with potential function class $\Phi$.*

*Proof.* Intuitively, this is possible since we can iterate over possible values for the hub node 0, and once the hub node is fixed, the graphical model reduces to $\sqrt{n}$ independent Markov chains, for which MAP estimation is tractable via dynamic programming. We proceed to the formal proof.

Fix any $J \in \Phi^E$. As preliminary notation, for each $c, c_0 \in \{0,1\}$ and $i, j \in \sqrt{n}$, let $V(i,j) := \{0\} \cup \{(k,j) : 1 \leq k \leq i\}$, and let $E(i,j)$ be the edge set of the induced subgraph $G[V(i,j)]$. For all indices $i, j \in \sqrt{n}$ and values

$c, c_0 \in \{0,1\}$, let $\mathcal{X}(c_0, c; i, j)$ denote the set of all partial configurations $x \in \{0,1\}^{V(i,j)}$ satisfying the "boundary conditions" $x_0 = c_0$ and $x_{(i,j)} = c$. With this notation, define

$$\hat{x}_{i,j}(c, c_0; J) := \underset{x \in \mathcal{X}(c_0, c; i, j)}{\arg\min} \sum_{(a,b) \in E(i,j)} J_{\{a,b\}}(x_a, x_b),$$

$$\hat{C}_{i,j}(c, c_0; J) := \underset{x \in \mathcal{X}(c_0, c; i, j)}{\min} \sum_{(a,b) \in E(i,j)} J_{\{a,b\}}(x_a, x_b).$$

For each $j \in [\sqrt{n}]$, let

$$\hat{x}_j(c_0; J) := \hat{x}_{\sqrt{n}, j}\left(\left(\underset{c \in \{0,1\}}{\arg\min} \hat{C}_{\sqrt{n}, j}(c, c_0; J)\right), c_0; J\right).$$

Finally, let $\hat{x}(c_0; J) \in \{0,1\}^V$ be the vector which takes value $c_0$ on vertex $0$, and value $\hat{x}_j(c_0; J)_i$ on vertex $(i, j)$ for all $i, j \in \sqrt{n}$. Let

$$\hat{x}(J) := \underset{c_0 \in \{0,1\}}{\arg\max} \, p_J(\hat{x}(c_0; J)).$$

We claim that $\hat{x}(J)$ is a maximizer of $p_J(x)$. Indeed, for any fixed $c_0 \in \{0,1\}$, $\hat{x}(c_0; J)$ is a maximizer of $p_J(x)$ subject to $x_0 = c_0$, because under this constraint the maximization problem decomposes into $\sqrt{n}$ independent maximization problems, one for each path in $G$, which by definition are solved by $\hat{x}_1(c_0; J), \ldots, \hat{x}_{\sqrt{n}}(c_0; J)$.

Moreover, it's straightforward to see that for any fixed $j$, $\hat{C}_j(c_0; J)$ can be computed in $O(\sqrt{n})$ time by dynamic programming. Indeed for any $i, j$, $\hat{C}_{i,j}(c, c_0; J)$ can be computed in $O(1)$ time from $\hat{C}_{i-1,j}(0, c_0; J)$ and $\hat{C}_{i-1,j}(1, c_0; J)$ as well as $J_{\{0,(i,j)\}}$ and $J_{\{(i-1,j),(i,j)\}}$. Once the values $\hat{C}_{i,j}(c, c_0; J)$ have been computed for all $i \in [\sqrt{n}]$ and $c \in \{0,1\}$, the vector $\hat{x}_j(c_0; J)$ can be computed in $O(\sqrt{n})$ time via a reverse scan over $i = \sqrt{n}, \ldots, 1$. It follows that $\hat{x}(J)$ can be computed in $O(n)$ time. $\square$

*Proof of Proposition 3.* We claim that there is a node message-passing protocol $P'$ on $G$ with $T + 1$ rounds that at each time $t \in [T + 1]$ has computed

$$P'_t(v; I) = (P_{t-1}(e; I))_{e \in \mathcal{M}(v)}.$$

That is, each node "simulates" the computation of all incident edges. The key point is that for any edge $e = (u, v) \in E$, the neighborhood $\mathcal{M}(e)$ is equal to $\mathcal{M}(u) \cup \mathcal{M}(v)$, so node $v$ can simulate the computation at $e$ using its own data and appropriate data from node $u$.

We make this idea formal by arguing inductively. Since $P_0 \equiv 0$, it's clear that this can be achieved for $t = 1$. Fix any $t > 1$ and suppose that $P'_{t-1}(u; I) = (P_{t-2}(e; I))_{e \in \mathcal{M}(u)}$

for all $u \in V$ and inputs $I$. For each $v \in V$, we define a function $f'_{t,v}$ by

$$f'_{t,v}((c(v'))_{v' \in \mathcal{N}(v)}, (I(e))_{e \in \mathcal{M}(v)})_{e^\star}$$
$$:= f_{t-1,e^\star}((c(v)_e)_{e \in \mathcal{M}(v)}, (c(v^\star)_e)_{e \in \mathcal{M}(v^\star)}, I(e^\star))$$

for each $e^\star = (v, v^\star) \in \mathcal{M}(v)$. Then by definition and the inductive hypothesis, we have

$$P'_t(v; I)_{e^\star}$$
$$= f'_{t,v}((P'_{t-1}(v'; I))_{v' \in \mathcal{N}(v)}, (I(e))_{e \in \mathcal{M}(v)})_{e^\star}$$
$$= f_{t-1,e^\star}((P'_{t-1}(v; I)_e)_{e \in \mathcal{M}(v)},$$
$$\quad (P'_{t-1}(v^\star; I)_e)_{e \in \mathcal{M}(v^\star)}, I(e^\star))$$
$$= f_{t-1,e^\star}((P_{t-2}(e; I))_{e \in \mathcal{M}(v)},$$
$$\quad (P_{t-2}(e; I)_e)_{e \in \mathcal{M}(v^\star)}, I(e^\star))$$
$$= P_{t-1}(e^\star; I)$$

for any edge $e^\star = (v, v^\star) \in E$, since $\mathcal{M}(e) = \mathcal{M}(v) \cup \mathcal{M}(v^\star)$. This completes the induction and shows that $P'_{T+1}(v; I) = (P_T(e; I))_{e \in \mathcal{M}(v)}$ for all $v, I$. Replacing $f'_{T+1,v}$ by $\tilde{f}_{T,v} \circ f'_{T+1,v}$ completes the proof. $\square$

## C. Omitted Proofs from Section 6

In this section we provide a formal proof of Theorem 4. For notational convenience, define $m = \lfloor \sqrt{n} \rfloor$. We define a graph $G = (V, E)$ that is a perfect $n$-ary tree of depth two. Formally, the graph $G$ has vertex set $V = \{0\} \cup [m] \cup ([m] \times [m])$. Vertex $0$ is adjacent to each $i \in [m]$, and each $i \in [m]$ is additionally adjacent to $(i, j)$ for all $j \in [m]$. We define a function $g : \{0,1\}^E \to \{0,1\}^V$ as follows. On input $I \in \{0,1\}^E$, for each edge $e \in E$, define the *input summation* at $e$ to be

$$C(I)_e := \sum_{e' \in \mathcal{M}(e)} I(e').$$

Intuitively, one may think of $C(I)_e$ as simulating the input on $e$ in the "large alphabet" construction described in Section 6. Next, define

$$g(I)_{(u,j)} := 0.$$
$$g(I)_u := \mathbb{1}[\#|e \in \mathcal{M}(\{0, u\}) :$$
$$\quad C(I)_e = C(I)_{\{0,u\}}| > m + 1].$$
$$g(I)_0 := \mathbb{1}[\exists u \in [m] : g(I)_u = 1].$$

In words, $g(I)_u$ is the indicator for the event that, among the $2m+1$ edges adjacent to $\{0, u\}$ (which include $\{0, u\}$ itself), more than $m + 1$ edges have the same input summation as $\{0, u\}$. At a high level, this definition of $g$ was designed

to satisfy three criteria. First, $g(I)_u$ depends on the input values on other branches of the tree: in particular, if $I_{\{0,v\}} = 0$ for all $v \in [n]$, then $C(I)_e = C(I)_{\{0,u\}}$ for all edges $e$ in the subtree of $u$, so $g(I)_u$ exactly measures the event that there is *at least one* edge $e$ outside the subtree of $u$ for which $C(I)_e = C(I)_{\{0,u\}}$. Second, there is no concise "summary" of $I$ such that $g(I)_u$ can be determined from this summary in conjunction with the inputs on the subtree of $u$. Third, $g(I)$ is equivariant to re-labelings of the tree.

The first two criteria, together with the fact that the root vertex $0$ is an "information bottleneck" for $G$, can be used to show that any node message-passing algorithm that computes $g$ on $G$ requires either large memory or many rounds. The third criterion enables construction of a symmetric edge message-passing protocol for $g$. The arguments are formalized in the claims below.

**Claim 7.** *For graph $G$ and function $g$ as defined above, any node message-passing protocol on $G$ that computes $g$ with $T$ rounds and $B$ bits of memory requires $TB \geq \Omega(m)$.*

*Proof.* Consider any input $I \in \{0,1\}^E$ with $I(\{0,u\}) = 0$ for all $u \in [m]$. Then for any $u, j \in [m]$, we have

$$C(I)_{\{u,(u,j)\}} = C(I)_{\{0,u\}} = \sum_{i=1}^{m} I(\{u,(u,i)\}).$$

Thus $g(I)_u = 1$ if and only if there exists some $v \in [m] \setminus \{u\}$ with $C(I)_{\{0,u\}} = C(I)_{\{0,v\}}$, or equivalently $\sum_{i=1}^{m} I(\{u,(u,i)\}) = \sum_{i=1}^{m} I(\{v,(v,i)\})$.

Fix $T, B$ and suppose that $P$ is a node message-passing protocol on $G$ that computes $g$ with $T$ rounds and $B$ bits of memory. Define sets of vertices $K := \{0\}$ and $S := \{1, \ldots, m/2\}$. Let $H := G[\overline{K}]$ and $F := \mathcal{M}(N_H^{T-1}(S))$. Then for any $T$, we have that

$$F = \{\{0,u\} : 1 \leq u \leq m/2\}$$
$$\cup \{\{u,(u,j)\} : 1 \leq u \leq m/2, 1 \leq j \leq m\}.$$

Define a vector $I_F \in \Phi^F$ by

$$I_{\{0,u\}} = 0 \text{ for } 1 \leq u \leq m/2$$
$$I_{\{u,(u,j)\}} = \mathbb{1}[j \leq u] \text{ for } 1 \leq u \leq m/2, 1 \leq j \leq m.$$

Now fix any $x \in \{0,1\}^S$. We claim that there is some $I_{\overline{F}} \in \Phi^{\overline{F}}$ such that $g_S(I_F, I_{\overline{F}}) = x$. Indeed, let us define $I_{\overline{F}}$ by:

$$I_{\{0,v\}} = 0 \quad \text{for } m/2 < v \leq m$$
$$I_{\{v,(v,j)\}} = x_{v-m/2}\mathbb{1}[j \leq v - m/2]$$
$$\text{for } m/2 < v \leq m, 1 \leq j \leq m.$$

Then $C(I)_{\{0,u\}} = u$ for all $1 \leq u \leq m/2$, and $C(I)_{\{0,v\}} = (v - m/2)x_{v-m/2}$ for all $m/2 < v \leq m$. It follows that for any $1 \leq u \leq n/2$, $x_u = 1$ if and only if there exists some $v \in [m] \setminus u$ with $C(I)_{\{0,u\}} = C(I)_{\{0,v\}}$, and hence $x_u = g(I)_u$. We conclude that

$$\left| \left\{ g_S(I_F, I_{\overline{F}}) : I_{\overline{F}} \in \Phi^{\overline{F}} \right\} \right| \geq 2^{m/2}.$$

Applying Lemma 2 we conclude that $TB \geq \Omega(m)$ as claimed. $\square$

**Claim 8.** *For graph $G$ and function $g$ as defined above, there is a symmetric edge message-passing protocol on $G$ that computes $g$ with $O(1)$ rounds and $O(\log m)$ bits of memory.*

*Proof.* In the first round, each edge processor reads its input value. In the second round, each edge processor sums the values computed by all neighboring edges (including itself). In the third round, each edge processor computes the indicator for the event that strictly more than $m + 1$ neighboring edges (including itself) have the same value as itself. In the final aggregation round, the output of a vertex is the indicator for the event that any neighbor has value 1.

By construction, the value computed by any edge $e$ after the second round is exactly $C(I)_e$. Thus, after the third round, the value computed by any edge $\{0,u\}$ is exactly $g(I)_u$. Moreover, the value computed by any edge $\{u,(u,j)\}$ is $0$ after the third round, since such edges only have $m + 1$ neighbors. It follows by construction of the final aggregation step that the protocol computes $g$. $\square$

*Proof of Theorem 4.* Immediate from Claims 7 and 8. $\square$

# D. Omitted Proofs from Section 7

*Proof of Theorem 5.* Without loss of generality, we may assume that the functions $(f_t^{\mathsf{sym}})_{t \in [T]}$ and $\tilde{f}^{\mathsf{sym}}$ are all the identity function (on the appropriate domains). The reason is that any symmetric edge message-passing protocol $\tilde{P}$ on $T$ rounds may be simulated by running $P$ and then applying a universal function (depending only on $\tilde{P}$) to each node's output value – see Lemma 9.

We argue by induction that for each $t \in [T]$, there is a $(t + 1)$-round symmetric node message-passing protocol that, on any input $I$, computes the function $Q_t(u; I) := \{\!\{P_t(e; I) : e \in \mathcal{M}(u)\}\!\}$ for every node $u \in V$. That is, the protocol at node $u$ simulates the *multiset* of computations performed by edges incident to $u$. This is similar to the idea for Proposition 3 but requires significant care to ensure symmetry of the protocol is preserved.

Consider $t = 1$. For any $e = (u, v) \in E$, we have by symmetry and the initial assumption that

$$P_1(e; I) = (I(e), 0, \{\!\{\{\!\{0 : v' \in \mathcal{N}(u)\}\!\}, \{\!\{0 : u' \in \mathcal{N}(v)\}\!\}\}\!\}). \tag{6}$$

We define a two-round node message-passing protocol on $G$ where the first update at node $u$ computes

$$P_1'(u; I) = \{\!\{I(e) : e \in \mathcal{M}(u)\}\!\}.$$

For the second update at node $u$, the node is required to compute a function of the data $(P_1'(u; I), \{\!\{(P_1'(v; I), I(\{u, v\})) : v \in \mathcal{N}(u)\}\!\})$. It does so by applying the following transformation to this data:

$$
\begin{aligned}
&(P_1'(u; I), \{\!\{(P_1'(v; I), I(\{u, v\})) : v \in \mathcal{N}(u)\}\!\}) \\
&\mapsto \{\!\{(I(\{u, v\}), 0, |\mathcal{N}(u)|, |P_1'(v; I)|) : v \in \mathcal{N}(u)\}\!\} \\
&\mapsto \{\!\{(I(\{u, v\}), 0, \{\!\{|\mathcal{N}(u)|, |P_1'(v; I)|\}\!\}) : v \in \mathcal{N}(u)\}\!\} \\
&= \{\!\{P_1(\{u, v\}; I) : v \in \mathcal{N}(u)\}\!\} =: P_2'(u; I),
\end{aligned}
$$

where the first step drops the term $P_1'(u; I)$, inserts the constant $|\mathcal{N}(u)|$ into each element of the multiset, and replaces each set $P_1'(v; I)$ by its cardinality; the second step symmetrizes the tuple $(|\mathcal{N}(u)|, |P_1'(v; I)|)$; and the equality uses the fact that $|P_1'(v; I)| = |\mathcal{N}(v)|$ together with Eq. (6). By construction, this protocol is symmetric, and we can see that $P_2'(u; I) = Q_t(u; I)$, which proves the induction for step $t = 1$.

Now pick any $t > 1$. For any $e = \{u, v\} \in E$, we know that the original protocol's computation at $e$ can be written as:

$$P_t(e; I) = (I(e), P_{t-1}(e; I), \{\!\{Q_{t-1}(u; I), Q_{t-1}(v; I)\}\!\}).$$

By the induction hypothesis, there is a $t$-round symmetric node message-passing protocol $P'$ that, at node $v$ on input $I$, computes

$$P_t'(v; I) = \{\!\{P_{t-1}(e; I) : e \in \mathcal{M}(v)\}\!\} = Q_{t-1}(v; I).$$

Note that since $P_{t-1}(e; I)$ is an element of the tuple $P_t(e; I)$, for each $1 \leq s \leq t - 1$ there is a fixed function $\gamma_s$ such that $\gamma_s(Q_{t-1}(v; I)) = Q_s(v; I)$ for all $v, I$. Using this fact, we extend $P'$ to $t + 1$ rounds. The update at round $t + 1$ and node $u$ is required to be a function of

the data $(P_t'(u; I), \{\!\{(P_t'(v; I), I(\{u, v\})) : v \in \mathcal{N}(u)\}\!\})$. By the induction hypothesis, this is equal to the data $(Q_{t-1}(u; I), \{\!\{(Q_{t-1}(v; I), I(\{u, v\})) : v \in \mathcal{N}(u)\}\!\})$. For notational convenience, write

$$S_{t-1, u} := \{\!\{(Q_{t-1}(v; I), I(\{u, v\})) : v \in \mathcal{N}(u)\}\!\}$$

and

$$S_{1:t-1, u} := \{\!\{(Q_{1:t-1}(v; I), I(\{u, v\})) : v \in \mathcal{N}(u)\}\!\}$$

where $Q_{1:t-1}(u; I)$ refers to the tuple $(Q_1(u; I), \ldots, Q_{t-1}(u; I))$. Observe that $Q_{1:t-1}(v; I)$ can be determined from $Q_{t-1}(v; I)$ (for any $v$) due to the existence of the functions $\gamma_1, \ldots, \gamma_{t-1}$; hence, $S_{1:t-1, u}$ can be computed from $S_{t-1, u}$. Using these observations, defining $P_{t+1}'(u; I)$ via the following sequence of transformations to the data $(Q_{t-1}(u; I), S_{t-1, u})$ is well-defined:

$$
\begin{aligned}
&(Q_{t-1}(u; I), S_{t-1, u}) \\
&\mapsto (Q_{1:t-1}(u; I), S_{1:t-1, u}) \\
&\mapsto \{\!\{(I(\{u, v\}), \\
&\qquad \{\!\{Q_{1:t-1}(u; I), Q_{1:t-1}(v; I)\}\!\}) : v \in \mathcal{N}(u)\}\!\} \\
&\mapsto \{\!\{(I(\{u, v\}), P_{t-1}(\{u, v\}; I), \\
&\qquad \{\!\{Q_{t-1}(u; I), Q_{t-1}(v; I)\}\!\}) : v \in \mathcal{N}(u)\}\!\} \\
&= \{\!\{P_t(\{u, v\}; I) : v \in \mathcal{N}(u)\}\!\} \\
&= Q_t(u; I) =: P_{t+1}'(u; I)
\end{aligned}
$$

The second transformation inserts $Q_{1:t-1}(u; I)$ into each element of the multiset $S_{1:t-1}(u; I)$ and symmetrizes with $Q_{1:t-1}(v; I)$. The final transformation drops $Q_{1:t-2}(u; I)$ and $Q_{1:t-2}(v; I)$ and inserts $P_{t-1}(\{u, v\}; I)$. This insertion is well-defined because the definition of $P_{t-1}(\{u, v\}; I)$ can be iteratively unpacked, and it is ultimately a function of the existing data

$$(I(\{u, v\}), \{\!\{Q_{1:t-1}(u; I), Q_{1:t-1}(v; I)\}\!\}).$$

To conclude, we have shown that $P'$ computes $Q_t(v; I)$ at node $u$ on input $I$, and that this can be achieved while satisfying symmetry. This completes the induction. Since $Q_T(u; I)$ is precisely the output of $P$ at node $u$ on input $I$ (after the node aggregation step), this shows that $P$ can be simulated by a $(T + 1)$-round symmetric node message-passing protocol on $G$. $\qquad\square$

**Lemma 9.** *Let* $T \geq 1$*, and let* $P = ((f_{t,e})_{t \in [T], e \in E}, (\tilde{f}_v)_{v \in V})$ *be a symmetric edge message-passing protocol on* $G = (V, E)$ *with* $T$ *rounds. Consider the* $T$*-round edge message-passing protocol* $P^\circ = ((f_{t,e}^\circ)_{t \in [T], e \in E}, (\tilde{f}_v^\circ)_{v \in V})$ *where for all* $t, e$*,*

$$
\begin{aligned}
&f_{t,e}^\circ((c(e'))_{e' \in \mathcal{M}(e)}, I(e)) \\
&\quad := (I(e), c(e), \{\!\{c(e') : e' \in \mathcal{M}(u)\}\!\}, \\
&\qquad \{\!\{c(e') : e' \in \mathcal{M}(v)\}\!\}),
\end{aligned}
$$

*and for every $v \in V$,*

$$\tilde{f}_v^\circ((c(e))_{e \in \mathcal{M}(v)}) := \{\!\{c(e) : e \in \mathcal{M}(v)\}\!\}.$$

*Then there is a function $h$ such that $\tilde{f}_v((P_T(e; I))_{e \in \mathcal{M}(v)}) = h(\tilde{f}_v^\circ((P_T^\circ(e; I))_{e \in \mathcal{M}(v)}))$ for all $v, I$.*

*Proof.* We prove by induction that for each $t \in \{0, \ldots, T\}$ there is a function $h_t$ such that $P_t(e; I) = h_t(P_t^\circ(e; I))$ for all $e, I$. For $t = 0$ this is immediate from the convention that $P_0 \equiv P_0^\circ \equiv 0$. Fix any $t \in \{1, \ldots, T\}$. Since $P$ is symmetric, there is a function $f_t^{\text{sym}}$ so that for all $e = (u, v) \in E$ and inputs $I$,

$$P_t(e; I) = f_t^{\text{sym}}(I(e), P_{t-1}(e; I), \{\!\{P_{t-1}(e'; I) : e' \in \mathcal{M}(u)\}\!\},$$
$$\{\!\{P_{t-1}(e'; I) : e' \in \mathcal{M}(v)\}\!\})$$
$$= f_t^{\text{sym}}(I(e), h_{t-1}(P_{t-1}^\circ(e; I)),$$
$$\{\!\{h_{t-1}(P_{t-1}^\circ(e'; I)) : e' \in \mathcal{M}(u)\}\!\},$$
$$\{\!\{h_{t-1}(P_{t-1}^\circ(e'; I)) : e' \in \mathcal{M}(v)\}\!\})$$

which is indeed a well-defined function (independent of $e, I$) of

$$P_t^\circ(e; I) = (I(e), P_{t-1}^\circ(e; I), \{\!\{P_{t-1}^\circ(e'; I) : e' \in \mathcal{M}(u)\}\!\},$$
$$\{\!\{P_{t-1}^\circ(e'; I) : e' \in \mathcal{M}(v)\}\!\}).$$

This completes the induction. Finally, since $P$ is symmetric, there is a function $\tilde{f}^{\text{sym}}$ such that $\tilde{f}_v((P_T(e; I))_{e \in \mathcal{M}(v)}) = \tilde{f}^{\text{sym}}(\{\!\{P_T(e; I) : e \in \mathcal{M}(v)\}\!\})$ for all $v, I$. Hence we can write

$$\tilde{f}_v((P_T(e; I))_{e \in \mathcal{M}(v)}) = \tilde{f}^{\text{sym}}(\{\!\{P_T(e; I) : e \in \mathcal{M}(v)\}\!\})$$
$$= \tilde{f}^{\text{sym}}(\{\!\{h_T(P_T^\circ(e; I)) : e \in \mathcal{M}(v)\}\!\})$$

which is a well-defined function (independent of $v, I$) of $\{\!\{P_T^\circ(e; I) : e \in \mathcal{M}(v)\}\!\}$ as needed. $\square$

## E. A quantitatively tight depth/memory separation

For each $n \in \mathbb{N}$, let $K_n := ([n], E_n)$ be the complete graph on $[n]$. In this section we show that there is a function that can be computed by an edge message-passing protocol on $K_n$ with constant rounds and constant memory per processor, but for which any node message-passing protocol with $T$ rounds and $B$ bits of memory requires $TB \geq \Omega(n)$. We remark that this separation is quantitatively tight due to Proposition 3, although it is possible that a larger (e.g. even super-polynomial in $n$) depth separation may be possible if

the node message-passing protocol is restricted to constant memory per processor.

At a technical level, the lower bound proceeds via a reduction from the *set disjointness problem* in communication complexity, similar to the lower bounds in (Loukas, 2020a).

**Definition 12.** Fix $m \in \mathbb{N}$. The set disjointness function $\mathsf{DISJ}_m : \{0, 1\}^m \times \{0, 1\}^m \to \{0, 1\}$ is defined as

$$\mathsf{DISJ}_m(A, B) := \mathbb{1}[\forall i \in [m] : A_i B_i = 0].$$

The following fact is well-known; see e.g. discussion in (Håstad & Wigderson, 2007).

**Lemma 10.** *In the two-party deterministic communication model, the deterministic communication complexity of $\mathsf{DISJ}_m$ is at least $m$.*

The main result of this section is the following:

**Theorem 11.** *Fix any even $n \in \mathbb{N}$. Define $g : \{0, 1\}^{E_n} \to \{0, 1\}^n$ by*

$$g(I)_v := \mathbb{1}\big[\exists\{i, j\} \in E_n : i, j \leq n/2$$
$$\wedge\, I(\{i, j\}) = I(\{n + 1 - i, n + 1 - j\}) = 1\big]$$

*for all $I \in \{0, 1\}^{E_n}$ and $v \in [n]$. Then the following properties hold:*

- *Any node message-passing protocol on $K_n$ with $T$ rounds and $B$ bits of memory that computes $g$ requires $TB \geq \Omega(n)$*

- *There is an edge message-passing protocol on $K_n$ with $O(1)$ rounds and $O(1)$ bits of memory that computes $g$.*

*Proof.* Let $m := \binom{n/2}{2}$. Let $P = (f_{t,v})_{t,v}$ be a node message-passing protocol on $K_n$ that computes $g$ with $T$ rounds and $B$ bits of memory. We design a two-party communication protocol for $\mathsf{DISJ}_m$ as follows. Suppose that Alice holds input $X \in \{0, 1\}^m$ and Bob holds input $Y \in \{0, 1\}^m$. Let us index the edges $\{i, j\} \in E_n$ with $i, j \leq n/2$ by $[m]$, and similarly index the edges $\{i, j\} \in E_n$ with $i, j > n/2$ by $[m]$, in such a way that edge $\{i, j\}$ has the same index as edge $\{n + 1 - i, n + 1 - j\}$. Let $I \in \{0, 1\}^{E_n}$ be defined by

$$I(\{i, j\}) := \begin{cases} X_{\{i,j\}} & \text{if } i, j \leq n/2 \\ Y_{\{i,j\}} & \text{if } i, j > n/2 \\ 0 & \text{otherwise} \end{cases}.$$

Initially, Alice computes $\hat{P}_0(v) := 0$ for all $v \in \{1, \dots, n/2\}$, and Bob computes $\hat{P}_0(v) := 0$ for all $v \in \{n/2 + 1, \dots, n\}$. The communication protocol then proceeds in $T$ rounds. At round $t \in [T]$, Alice sends $(\hat{P}_{t-1}(v))_{1 \le v \le n/2}$ to Bob, and Bob sends $(\hat{P}_{t-1}(v))_{n/2+1 \le v \le n}$ to Alice. Alice then computes

$$\hat{P}_t(v) := f_{t,v}((\hat{P}_{t-1}(v'))_{v' \in [n]}, (I(e))_{e \in M_{K_n}(v)})$$

for each $1 \le v \le n/2$, and Bob computes the same for each $n/2 < v \le n$. Note that for any $i \le n/2$ and edge $e \in M_{K_n}(i)$, Alice can compute $I(e)$. Similarly, for any $i > n/2$ and edge $e \in M_{K_n}(i)$, Bob can compute $I(e)$. Thus, this computation is well-defined. After round $T$, Alice and Bob output $1 - \hat{P}_T(1)$ and $1 - \hat{P}_T(n)$ respectively.

This defines a communication protocol. Since $\hat{P}_t(v) \in \{0,1\}^B$ for each $v \in [n]$ and $t \in [T]$, the total number of bits communicated is at most $nBT$. Moreover, by induction it's clear that Alice and Bob output $1 - P_T(1; I)$ and $1 - P_T(n; I)$ respectively. By assumption that $P$ computes $g$ and the fact that $g(I)_v = 1 - \mathsf{DISJ}_m(X, Y)$ for all $v \in [n]$, we have that $1 - P_T(1; I) = 1 - P_T(n; I) = 0$ if $\mathsf{DISJ}_m(I) = 0$, and $1 - P_T(1; I) = 1 - P_T(n; I) = 1$ if $\mathsf{DISJ}_m(I) = 1$. Thus, this communication protocol computes $\mathsf{DISJ}_m$. By Lemma 10, it follows that $nBT \ge m = \Omega(n^2)$, so $BT = \Omega(n)$ as claimed.

Next, we exhibit an edge message-passing protocol on $K_n$ that computes $g$ with six rounds and one bit of memory. Intuitively, the protocol proceeds via the following steps:

1. First, each edge "reads" its input.

2. Second, each edge $\{i, j\}$ swaps its value with the value at $\{n + 1 - i, n + 1 - j\}$; since these two edges are not adjacent, this takes two steps.

3. Third, each edge $\{i, j\}$ with $i, j \le n/2$ checks if the input at $\{n + 1 - i, n + 1 - j\}$ (which it now knows) equals its own input.

4. Fourth, an aggregation step is performed across the entire graph. Since the graph is complete, this can be done in two steps.

We proceed to make this intuition more formal. For $1 \le t \le 6$ and $e \in E_n$, define $f_{t,e} : \{0,1\}^{\mathcal{M}(e)} \times \{0,1\} \to \{0,1\}$ as follows:

$$f_{1,\{i,j\}}(x, y) := y$$
$$f_{2,\{i,j\}}(x, y) := x_{\{n+1-i,j\}}$$
$$f_{3,\{i,j\}}(x, y) := x_{\{i,n+1-j\}}$$
$$f_{4,\{i,j\}}(x, y) := \mathbb{1}[y = x_{\{i,j\}} \wedge i, j \le n/2]$$
$$f_{5,\{i,j\}}(x, y) := \mathbb{1}[\exists k \in [n] : x_{\{i,k\}} = 1]$$
$$f_{6,\{i,j\}}(x, y) := \mathbb{1}[\exists k \in [n] : x_{\{i,k\}} = 1].$$

Also define $\tilde{f}_v : \{0,1\}^{\mathcal{M}(v)} \to \{0,1\}$ for each $v \in [n]$ by $\tilde{f}_v(x) := x_{\{x,1\}}$. It can be checked that the computation of $P$ at timestep $t = 6$ is

$$P_6(\{i,j\}; I)$$
$$:= \mathbb{1}[\exists k, \ell \in [n/2] : I(\{k,\ell\}) = I(\{n+1-k, n+1-\ell\})]$$
$$= g(I).$$

From the definition of $\tilde{f}$, it follows that $P$ computes $g$. $\quad\square$

## F. Further details on synthetic task over Ising models

### F.1. Background on belief propagation

A classical way to calculate the marginals $\{\mathbb{E}[x_i]\}$ of an Ising model, when the associated graph is a tree, is to iterate the message passing algorithm:

$$\nu_{i \to j}^{(t+1)} = \tanh\left(h_i + \sum_{k \in \partial_i \setminus j} \tanh^{-1}\left(\tanh(J_{ik})\nu_{k \to i}^{(t)}\right)\right) \tag{7}$$

When the graph is a tree, it is a classical result ((Mezard & Montanari, 2009), Theorem 14.1) that the above message-passing algorithm converge to values $\nu^*$ that yield the correct marginals, namely:

$$\mathbb{E}[x_i] = \tanh\left(h_i + \sum_{k \in \partial_i} \tanh^{-1}\left(\tanh(J_{ik})\nu_{k \to i}^*\right)\right).$$

The reason the updates converge to the correct values on a tree topology is that they implicitly simulate a dynamic program. Namely, we can write down a recursive formula for the marginal of node $i$ which depends on sums spanning each of the subtrees of the neighbors of $i$ (i.e., for each neighbor $j$, the subgraph containing $j$ that we would get if we removed edge $\{i, j\}$).

If we root the tree at an arbitrary node $r$, we can see that after completing a round of message passing from the leaves to the root, and another from the root to the leaves, each subtree of $i$ will be (inductively) calculated correctly.

Moreover, even though the updates (7) are written over edges, the dynamic programming view makes it clear an equivalent message-passing scheme can be written down where states are maintained over the *nodes* in the graph. Namely, for each node $v$, we can maintain two values $h_{v,\text{down}}$ and $h_{v,\text{up}}$, which correspond to the values that will be used when $v$ sends a message upwards (towards the

root) or downwards (away from the root). Then, for appropriately defined functions $F, G$ (depending on the potentials $J$ and $h$), one can "simulate" the updates in (7):

$$h_{v,\text{up}}^{(t+1)} \leftarrow F\left(\{h_{w,\text{up}}^{(t)} : w \in v \cup \text{Children}(v)\}\right) \qquad (8)$$

$$h_{v,\text{down}}^{(t+1)} \leftarrow G\left(h_{\text{Parent}(v),\text{down}}^{(t)}, \left\{h_{w,\text{up}}^{(t)}\right\}_{w \in \text{Children}(v)}\right) \qquad (9)$$

Intuitively, $h_{v,\text{up}}$ captures the effective external field induced by the subtree rooted at $v$ on $\text{Parent}(v)$. After the upward messages propagate, the root $r$ can compute its correct marginal. Once $h_{\text{Parent}(v),\text{down}}$ is the correct marginal for $\text{Parent}(v)$ at some step, $h_{v,\text{down}}$ will be the correct marginal for $v$ at all subsequent steps.

### F.2. GCN-based architectures to calculate marginals

The belief-propagation updates (7) naturally fit the general edge-message passing paradigm from (2). In fact, they fit even more closely a "directed" version of the paradigm, in which each edge $\{i, j\}$ maintains two embeddings $h_{i \rightarrow j}, h_{j \rightarrow i}$, such that the embedding for direction $h_{i \rightarrow j}$ depends on the embeddings $\{h_{k \rightarrow i}\}_{\{k,i\} \in E}$. With this modification to the standard edge GCN architecture Eq. (4), it is straightforward to implement (7) with one layer, using a particular choice of activation functions and weight matrices $W$ (since, in particular, in our dataset all edge potentials $J_{i,j}$ are set to 1). Similarly, with a directed version of the node GCN architecture Eq. (3), where each node maintains an "up" embedding as well as a "down" embedding, it is straightforward to implement the "node-based" dynamic programming solution (8)-(9).

We call the architectures that do not maintain directionality Node-U and Edge-U (depending on whether they use a node-based or edge-based GCN). We call the "directed" architectures Node-D and Edge-D respectively. Since there are only initial node features (input as node potentials $\{h_i\}_{i \in}$), for the edge based architectures we initialize the edge features as a concatenation of the node features of the endpoints of the edge. The results we report for each architecture are the best over a sweep of depth $\in \{5, 10, 15, 20, 25, 30\}$ and width $\in \{10, 32, 64\}$.

### F.3. Edge-based models improve over node-based models

In Figure 3 we show the results for several tree topologies: a complete binary tree (of size 31), a path graph (of size 30), and uniformly randomly chosen trees of size 30 (the results in Figure 3 are averaged over 3 samples of tree).

The architectures in the legend (Node-U, Edge-U, Node-D, Edge-D) are based on a standard GCN, and detailed in Section F.2

We can see that for both the undirected and directed versions, adding edge embeddings improves performance. The improved performance of all directed versions compared to their undirected counterpart is not very surprising: the standard, undirected GCN architecture treats all neighbors symmetrically — hence, the directed versions can more easily simulate something akin to the belief propagation updates (7) as well as the node-based dynamic programming (8)-(9).

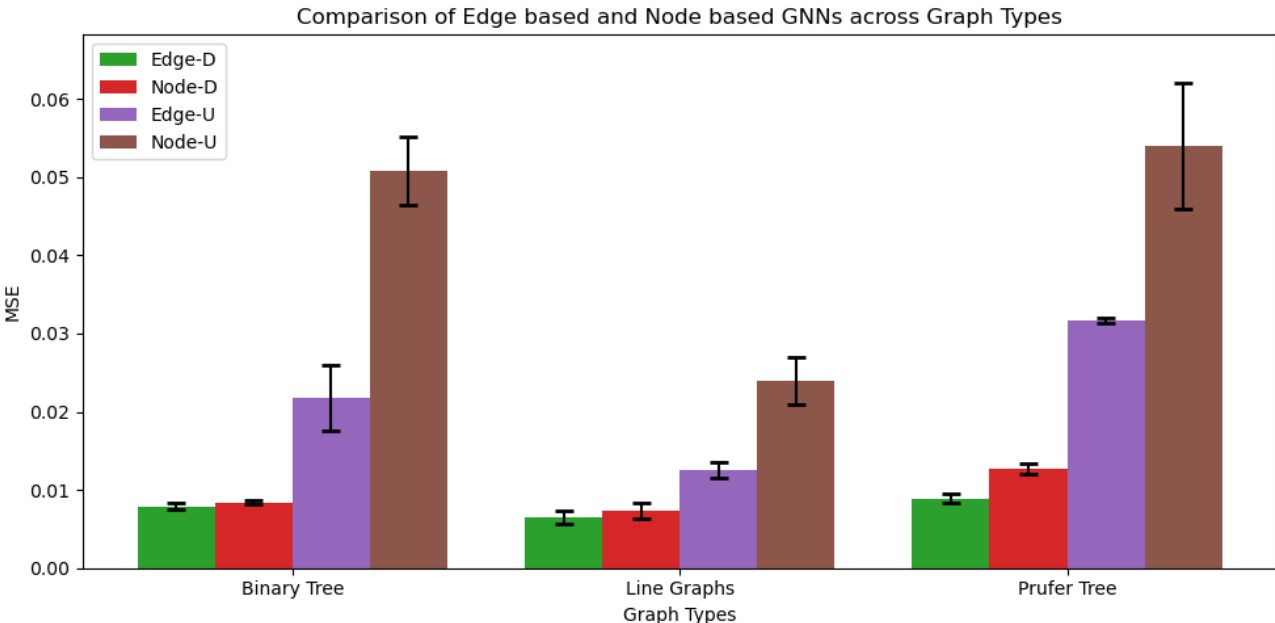

*Figure 3.* Comparison of four architectures for calculating node marginals in an Ising model. The architectures considered are node-embedding (3) and edge-embedding (4) versions of a GCN (correspondingly labeled Node-U and Edge-U), as well as their "directed" counterparts, as described in Section F.2, correspondingly labeled Node-D and Edge-D. The x-axis groups results according to the topology of the graph, the y-axis is MSE (lower is better). The mean and variances are reported over 3 runs for the best choice of depth and width over the sweep described in Section F.2.

