# OpenReview forum: "Towards characterizing the value of edge embeddings in Graph Neural Networks"
_ICML.cc/2025/Conference — ICML 2025 poster_

### Official Review · Reviewer_m8Mu · 2025-02-27

**Overall Recommendation:** 3

**Summary:**

*Updates after rebuttal: I have increased my score since my concern was addressed by the authors.*

———

This paper studies the benefits of using edge embeddings in graph neural network (GNN) as opposed to node embeddings. The authors theoretically show that  under memory constraints on the embeddings, an edge embedding GNN can solve certain graphical model task using a shallow model, whereas a node embedding GNN requires a much deeper model. The authors shows that such depth separation continues to hold with additional symmetry constraints (corresponding to commonly-used GNNs that satisfy permutation invariance). On the other hand, the authors prove that without memory constraint and using only symmetry constraints, there is no separation between edge embedding GNNs and node embedding GNNs. The theoretical findings on the benefits of edge embeddings GNNs are supported by empirical evidence on selected graph benchmarks and synthetic datasets.

**Claims And Evidence:**

Overall, the theoretical claims are mostly well supported. There are a few claims that required further explanations and/or revisions.

1. The main result on separation between node and edge message-passing protocols (Thm.1): while it is clear that there is a depth seperation (i.e. the number of message-passing rounds, or layers in GNN), it is unclear if this translates to difference in terms of total computational time. More concretely, Thm.1-2 shows that the edge protocol requires $O(1)$ rounds and $O(|\mathcal{M}(e)|)$ time. For star graphs, the centroid node of the star requires $O(n)$ time to evaluate the update rule, which suggests the total time is $O(n)$. On the other hand, the lower bound of node protocol shows $TB \ge \sqrt{n} - 1$, which suggests the total time is $\Omega(\sqrt{n})$. If so, then there seems to be no separation in terms of total computation time for this graphical model task? That said, I do appreciate the authors provide the tighter separation result using set disjointness in Appendix. E without the ambiguity of total compute time.

2. Remark 10: The authors claim that the $k$-Weisfeiler-Lehman (WL) test only characterizes the expressivity of higher-order GNNs with uninformative input features, which is not true. The graph can have informative node features and edge features, which will be utilized during the initialization step of $k$-WL, making the test more discriminative.

**Essential References Not Discussed:**

The following work investigates the power of using virtual node in node-based MPNNs.
[1]. Cai, Chen, et al. "On the connection between mpnn and graph transformer." International conference on machine learning. PMLR, 2023.

The following work discusses a mitigation strategy of over-squashing using graph rewiring.
[2]. Topping, Jake, et al. "Understanding over-squashing and bottlenecks on graphs via curvature." International Conference on Learning Representations.

**Experimental Designs Or Analyses:**

To provide more comprehensive empirical evidence for highlighting the theoretical insight, it will be nice to ablate on the number of layers when comparing the edge-based GNNs and node-based GNNs in Table 1.

**Methods And Evaluation Criteria:**

The authors demonstrated their theoretical insights on the benefits of edge embedding GNNs on synthetic graphs that are aligned with the theoretical construction. However, I am not sure the evaluation on some of the chosen real-world graphs is particularly meaningful, e.g. on MNIST and CIFAR-10 graphs; do the authors choose these benchmarks due to their topology resembling a star graph?

**Other Comments Or Suggestions:**

The notation $\Delta$ appears in Defn 6 (without explanation) and re-appears in Prop. 3 (with likely a different meaning). Consider fixing the possible notation clash.

**Other Strengths And Weaknesses:**

Strengths: The paper is well written overall, with interesting theoretical results on the depth separation between edge-based and node-based GNNs, using tools from communication theory and theoretical computer science. The theoretical insights are illustrated in both synthetic and real-world tasks.

Weakness:
1. As shown in prop.3 by the authors, the main depth separation result crucially relies on the existence of a high-degree vertex in the graph. As acknowledge by the authors, the edge-based GNNs often suffer from higher computational complexity. In light of these findings, the paper can be strengthened by proposing efficient approaches that close the separation between edge-based and node-based GNNs, such as graph rewriting in Topping et al.[2].
2. Some of the theoretical claims require further clarification/revision (see Claims and Evidence above).

**Questions For Authors:**

1. Does the depth separation result on MAP evaluator translate to time complexity separation? (See more details in Claims and Evidence)

2. Can the authors comment on the implications of their findings to the heuristic of virtual nodes in node-based GNNs? (see Relation to Literature)

3. Can the authors explain their choice of real-world graph benchmarks? (see Methods and Evaluation Criteria).

**Relation To Broader Scientific Literature:**

The comparison between edge-based GNN and node-based GNN is clearly of interest to the community, including both theoreticians and practitioners. The main theoretical results in showing the benefits of edge-based GNN mostly utilize the hub-node topology. This seems also  related to the practical heuristic in training GNNs for long-range task by adding virtual nodes (VN) (see more discussion and references in [1] Cai et al.). It will be nice for the authors to discuss their result in the context of virtual nodes.

**Theoretical Claims:**

I checked the proof of Thm.1: the overall argument seems correct but there are a few notation undefined. Specifically, in the statement of Lemma 2, the notations $\bar{K}, \bar{F}$ are undefined.

I also checked the correctness of the proof of Thm.4 and the proof sketch of Thm.5.

---

> ### Author Rebuttal · Authors · 2025-04-01
>
> We thank the reviewer for their thoughtful review and we appreciate the positive comments! Below, we address the reviewer’s concerns and questions:
>
> **Efficient approaches that close the separation between edge-based and node-based?** Thanks for pointing us to the reference by Topping et al. on graph rewiring; we’ll add it to the discussion. We agree that it would be nice to design more efficient approaches to close the separation. There are a variety of interesting heuristics for improving GNNs, though the trade-offs of such approaches are still not theoretically fully-understood. In particular, it would be very exciting to understand what representational effects (i.e. on required depth/width of a GNN) such heuristics have for specific choices of graphs and tasks.
>
> **Does the depth separation translate to time complexity separation (Question 1)?** Yes, there is a separation in parallel time complexity (with one processor per node/edge in the respective models), modulo an assumption that each processor reads all of its neighbors’ input at each round. In the edge model, the procedure can be implemented in time O(n) as the reviewer states. In the node model, we prove $TB \geq \sqrt{n}$. Since the hub node has $O(n)$ neighbors, each round requires parallel time at least $O(nB)$ if the hub node reads the $B$ bits for each neighbor, so the overall parallel time complexity is at least $O(nTB) = O(n^{3/2})$. We will clarify this point in Remark 4.
>
> **Remark about input features in WL test?** We agree that the k-WL refinement procedure can utilize informative features, and we apologize for the imprecise wording of Remark 10.
>
> In Remark 10 we were trying to briefly make a somewhat subtle point (which we will expand on): prior works on expressivity of GNNs (w.r.t. the WL test) measure expressivity by asking “for a particular input, what are the possible outputs” (and they show this is characterized by number of WL refinement steps). However, particularly for GNNs that take as input both a graph and informative input labels, we would argue the central representation question is what functions the GNN can represent, i.e. “what are the possible mappings from inputs to outputs”. This is what matters for downstream learning tasks, and what motivated our framework (see also the paragraph “GNNs as a computational machine” in Sec. 3). This is also analogous to the classical representational theory for standard neural networks.
>
> **Implication for virtual nodes (Question 2)?** Thanks for bringing up virtual nodes and the reference to Cai et al.; this is an interesting connection and we will add a remark to the paper. Since our main construction uses a node that is connected to the rest of the graph, it can be interpreted as showing a difficulty for memory-constrained virtual nodes (or even a motivation for designing GNNs with larger memory at the virtual node, which perhaps could be implemented by variable dimensionality of the embeddings maintained at different nodes in the GNN).
>
> **Choice of real-world benchmarks (Question 3)?** We tried to choose diverse tasks that enable fair comparison of edge-based vs. node-based models (thus, we ruled out edge classification and node classification tasks, because they introduce a confounder – they require substantively different collation layers between the two architectures). ZINC/MNIST/CIFAR-10 are graph classification/regression datasets from the original GNN benchmark of Dwivedi et al. [1], and Peptides-Func/Peptides-Struct are the graph classification/regression datasets from the long-range graph benchmark [2].
>
> Most of these graphs in fact do not have as skewed degree distributions as the synthetic examples (e.g. the chemistry graphs are subject to physical/molecular constraints, and the image graphs are constructed so that all degrees are within a factor of two – see Appendix C.5 in [1]). The goal of Table 1 was primarily to perform a controlled comparison between edge and node architectures on commonly-used GNN benchmark datasets. Of course, expanding the range of datasets (or even constructing more challenging ones reflective of natural GNN tasks) and exploring the impact of e.g. degree statistics on performance is an interesting future direction.
>
> **Notation in Lem. 2 / Def. 6 / Prop. 3?** The overline notation in Lem. 2 means set complement, and the Delta({0,1}^V) in Def. 6 means the set of distributions over {0,1}^V. We will clarify these (and Prop. 3).
>
> **Ablation on the number of layers?** We used the same number of layers for both architectures in Table 1 so as to equalize everything except the design choice “node-based vs edge-based”. Anecdotally we did not find significant changes with deeper node architectures, but unfortunately doing a thorough sweep over depths for both architectures would have been computationally taxing.
>
> [1] Dwivedi et al. “Benchmarking Graph Neural Networks”. JMLR, 2023.
>
> [2] Dwivedi et al. “Long Range Graph Benchmark”. NeurIPS 2022.

---

### Official Review · Reviewer_CZSf · 2025-03-07

**Overall Recommendation:** 3

**Summary:**

The paper focuses on message-passing that also consider edge embeddings. The authors show theoretically that edge embeddings can have substantial benefits in terms of how deep a model needs to be and run some experiments to verify this claim.

**Claims And Evidence:**

While the contributions of the work are mainly theoretical, Table 2 aims to provide support for the empirical claims.

**Essential References Not Discussed:**

I would not say this is essential, but in remark 9 I believe the info-theory and over-squashing could be better acknowledged, for example [1]

[1] Banerjee et al. Oversquashing in GNNs through the lens of information contraction and graph expansion

**Experimental Designs Or Analyses:**

Experimental section is rather limited, with Table 1 not contributing to the main claims very strongly in my opinion and Table 2 being a synthetic experiment.

**Methods And Evaluation Criteria:**

Table 1 includes mostly standard benchmarks. However I believe it is missing a key evaluation which is checking if the depth can be less for edge-GCN when compared to normal GCN. This is in fact seems to be one of the main claims: that maintaining edge information allows one to train a more shallow MPNN.

I also think that Table 2 might not be very surprising as it seems to me like the task heavily requires edge information, although I am only vaguely familiar with the Ising model.

**Other Comments Or Suggestions:**

N/A

**Other Strengths And Weaknesses:**

The paper tackles what I believe to be a relatively under-studied topic as people often ignore edge-embeddings. It is interesting to see this kind of work and the main intuitive results seem somewhat interesting. I am not sure that the main conclusion however of "adding edge embeddings provides additional power" is particularly illuminating.

**Questions For Authors:**

Could the authors provide more information on why they believe Table 1 supports the main claim that edge embeddings provide additional representational power? Should this not be an ablation involving depth of the model as this seems to be a main theoretical result?

**Relation To Broader Scientific Literature:**

As this paper is far from my area of expertise I am not completely sure. I found remark 9 on oversquashing interesting, there are indeed works that study the connection between oversquashing and information theory. I would have been interested on further comments on this connection.

**Theoretical Claims:**

I cannot comment on the proofs as they are quite far from my main area of research. As a consequence, I am not sure how illuminating they are.

---

> ### Author Rebuttal · Authors · 2025-04-01
>
> We thank the reviewer for their thoughtful review and we appreciate the positive comments! Below, we address the reviewer’s concerns and questions:
>
> **Interpretation of our main conclusions:** The reviewer is correct that it’s unsurprising that adding edge embeddings may provide additional (representational) power. However, the goal of our work is not just to answer this yes/no question, but rather to investigate *when* edge embeddings help, *how much* they help (in particular, what are instances in which edge embeddings are particularly helpful), and understand the *mechanism* by which they help. Our main theoretical conclusions are that:
>
> (1) adding edge embeddings (substantially) improves representational power in terms of required depth when there are *hub nodes*, and conversely does not improve representational power when the degree is bounded;
>
> (2) this phenomenon is a consequence of *memory constraints* (and in particular, their interplay with depth) and not present under the standard theoretical lens where the only constraint on the protocol is symmetry.
>
> We note these results can be viewed as GNN parallels of well-studied depth separation results for feedforward architectures [1] and Transformers [2] in which the goal is, similarly, to understand when and how much depth helps. These results were very influential in understanding the benefits of depth for classical architectures—but such theory is much less developed for GNNs.
>
> Our experimental results validate the “hub nodes” finding (Table 2) and demonstrate a noticeable but small gain on real-world benchmarks (Table 1). We hope that these results will enrich the conversation on *when* to use edge embeddings, motivate the search for real-world benchmarks with larger performance gaps (if they exist), and inspire the development of architectures that match the representational power of edge embeddings with better computational efficiency.
>
> We hope that this addresses the reviewer’s concern about whether our conclusions are “illuminating”.
>
> **Interpretation of Table 1:** We agree with the reviewer that the closest analogue of the theory would be an experiment of the form “edge-based architectures achieve the same accuracy with lower depth”. Table 1 is morally equivalent so long as one believes that higher depth improves accuracy: “edge-based architectures achieve higher accuracy with the same depth”. We chose the latter because sweeping over depths to match accuracies is computationally expensive (and, ultimately, accuracy subject to compute/size constraints is an important desideratum in its own right), but we agree that an experiment trying to determine some representational “thresholds” as a function of depth would be scientifically interesting.
>
> **Interpretation of Table 2:** Table 2 is actually about the planted model (section 8.2), not the Ising model – we apologize if the location of the table caused this confusion. Since the planted model is edge-based, it is obvious that the edge-based architecture is representationally powerful enough to solve the task, but Table 2 shows that the learning procedure also works (i.e. there are no unexpected training difficulties). Table 2 also shows that the node-based architecture empirically cannot learn (even though e.g. there is a node for every edge, since the graph is a star), consistent with our theoretical finding that hub nodes cause issues for node-based architectures.
>
> [1] Telgarsky, “Benefits of depth in neural networks”, COLT 2016.
>
> [2] Sanford et al. “Transformers, parallel computation, and logarithmic depth” ICML 2024.

---

> > ### Comment · Reviewer_CZSf · 2025-04-03
> >
> > Thanks for the response.
> >
> > After having read the responses to my questions and those of the other reviewers, I have decided to upgrade my score.

---

### Official Review · Reviewer_WSVk · 2025-03-14

**Overall Recommendation:** 4

**Summary:**

This paper studies how edge-based embeddings, rather than the more conventional node-based embeddings, can influence the representational power and performance of graph neural networks (GNNs). The authors formalize two message-passing models (one that maintains node embeddings, and another that maintains edge embeddings) and compare their ability to solve tasks under constraints on memory and depth.

**Claims And Evidence:**

The claims are well supported by rigorous theoretical arguments for which formal proofs are provided.  The authors also provided empirical evidence.

**Essential References Not Discussed:**

N/A

**Experimental Designs Or Analyses:**

The experiments on benchmark datasets apply standard training/test splits and well-accepted metrics (MAE, accuracy). This design is reasonable.

The authors also propose synthetic stress tests (star graphs and Ising trees). These are well motivated and precisely target the “hub bottleneck” phenomenon.

The experiments confirm the theoretical insight that hub-centered graphs show big performance gaps favoring edge embeddings.

**Methods And Evaluation Criteria:**

The experimental evaluation follows standard protocol and is adaquate. The authors clearly contrast edge-based vs. node-based architectures.

**Other Comments Or Suggestions:**

N/A

**Other Strengths And Weaknesses:**

Strengths:
- The paper provides both rigorous theoretical analysis and empirical support for its main assertion.
- The carefully designed synthetic experiments strongly highlight the advantages of edge-based GNNs in specific graph topologies.
- The authors’ presentation is generally clear, with explicit definitions (e.g., node vs. edge protocols) and clearly written proofs.

Weaknesses:
- In dense graphs, maintaining an embedding for each edge can become computationally expensive; the paper mentions this but could discuss more practical engineering considerations.

**Questions For Authors:**

see weaknesses

**Relation To Broader Scientific Literature:**

The related literature is accurately described and cited.

**Theoretical Claims:**

The main theoretical claims revolve around the existence of tasks (e.g., MAP inference on particular graphs) that require large depth for node-based but not for edge-based protocols, given constant local memory. Formal proofs are given an seem accurate to me — I did not identify any errors or issues in the line of argument.

---

> ### Author Rebuttal · Authors · 2025-04-01
>
> We thank the reviewer for their thoughtful review and we appreciate the positive comments!
>
> Regarding the *computational challenge posed by dense graphs*: we agree that mitigating this challenge while maintaining the representational power of edge-based architectures is an interesting direction for future work. A variety of heuristics have been proposed in the GNN literature to try to address related issues (e.g. graph rewiring, as mentioned by reviewer m8Mu). Developing a fuller theoretical understanding for these practical approaches is an exciting direction, and we are happy to add discussion to this effect in the paper.

---

### Official Review · Reviewer_1eR5 · 2025-03-14

**Overall Recommendation:** 4

**Summary:**

The authors explore when edge embeddings are more effective than the traditional node embeddings approaches in graph processing. Their theoretical findings suggest that node-based message passing struggles with certain tasks, especially under tight memory constraints, whereas edge processing offers a more efficient alternative in these cases. Interestingly, without memory limits, the two approaches are nearly equivalent. Experiments further support the benefits of edge embeddings, showing that they can enable shallower yet expressive architectures.

**Claims And Evidence:**

Yes

**Essential References Not Discussed:**

N/A

**Experimental Designs Or Analyses:**

N/A

**Methods And Evaluation Criteria:**

Yes

**Other Comments Or Suggestions:**

N/A

**Other Strengths And Weaknesses:**

Strengths:

[0] The authors provide a thorough analysis of edge embeddings in GNNs

[1] The writing is mostly clear and easy to follow

Weakness

[0] The performance gain in practice is quite small, though, in theory, there may exist graphs where the improvement is significant. However, it remains uncertain whether such graphs would naturally occur in real-world scenarios.

**Questions For Authors:**

In the real-world experiments provided, the edge-based GNN models' improvement over others appears to be marginal. I wonder whether the trade-off justifies the practical use of this approach?

**Relation To Broader Scientific Literature:**

This paper will be of significant interest to the graph machine learning community as it explores the applicability and trade-offs between edge-based and node-based embeddings, providing valuable theoretical insights.

**Theoretical Claims:**

I did not verify the correctness of all the theoretical proofs in appendix.

---

> ### Author Rebuttal · Authors · 2025-04-01
>
> We thank the reviewer for their thoughtful review and we appreciate the positive comments, particularly that the reviewer finds our work to be “of significant interest to the graph machine learning community”!
>
> Regarding the *size of the performance gain for edge-based GNNs on the real-world experiments*: we agree that the gain is marginal. However, we believe this is still a valuable experimental outcome. Note that edge-based GNNs are a pre-existing architecture with already numerous applications, and our goal in this paper was not to make an argument that they are inherently superior to node-based GNNs but rather to understand their benefits or drawbacks in controlled settings. A priori, edge-based architectures could have been much better or much worse than node-based architectures on existing benchmarks; Table 1 indicates that they are in fact slightly better.
>
> This outcome is consistent with our theoretical results since most of the graphs in these benchmarks do not have as skewed degree distributions as our theoretical constructions and synthetic examples (e.g. the chemistry graphs from ZINC/Peptides-func/Peptides-struct are subject to physical/molecular constraints, and the computer vision graphs from MNIST/CIFAR-10 are constructed in such a way that all degrees are within a factor of two – see Appendix C.5 in [1]). We agree completely that understanding whether there are naturally-arising “harder” benchmarks is an interesting direction for future research.
>
> [1] Dwivedi et al. “Benchmarking Graph Neural Networks”. JMLR, 2023.

---

> > ### Comment · Reviewer_1eR5 · 2025-04-02
> >
> > I thank the authors for their clarifications. Exploring edge embeddings through a theoretically rigorous lens, rather than following the conventional focus on node embeddings, is indeed a compelling direction. I will update my score accordingly.

---

### Decision · Program_Chairs · 2025-05-01

**Decision:**

Accept (poster)

**Comment:**

This paper studies how edge-based embeddings, rather than the more conventional node-based embeddings, can influence the representational power and performance of graph neural networks (GNNs). It is shown that certain tasks can be solved with a one layer GNN using edge representations and limited memory per feature, while they take $\Omega(\sqrt{n})$ layers for node-based GNNs with limited memory per feature. This shows that the memory lens, which is often overlooked by other forms of expressivity analysis, is an important aspect that affects expressivity. The paper also compares symmetry preserving and non-symmetric architectures. The results are corroborated by experiments.


All reviewers were positive about the paper, and agreed that the topic is timely and important, the paper is well written and clear, and the theoretical contribution is interesting, insightful, and rigorous. The reviewers also agreed, as stated also in the paper, that the main limitation of edge-based GNNs is that they often suffer from higher computational complexity with respect to node-based GNNs.


The authors are asked to implement the reviewer’s comments in the final version of the paper if accepted.